# Expression of the novel maternal centrosome assembly factor Wdr8 is required for vertebrate embryonic mitoses

Daigo Inoue[1], Manuel Stemmer[1], Thomas Thumberger[1], Thomas Ruppert[2], Felix Bärenz[2,†], Joachim Wittbrodt[1] & Oliver J. Gruss[2,3]

The assembly of the first centrosome occurs upon fertilisation when male centrioles recruit pericentriolar material (PCM) from the egg cytoplasm. The mechanisms underlying the proper assembly of centrosomes during early embryogenesis remain obscure. We identify Wdr8 as a novel maternally essential protein that is required for centrosome assembly during embryonic mitoses of medaka (*Oryzias latipes*). By CRISPR–Cas9-mediated knockout, maternal/zygotic *Wdr8*-null ($m/zWdr8^{-/-}$) blastomeres exhibit severe defects in centrosome structure that lead to asymmetric division, multipolar mitotic spindles and chromosome alignment errors. Via its WD40 domains, Wdr8 interacts with the centriolar satellite protein SSX2IP. Combining targeted gene knockout and *in vivo* reconstitution of the maternally essential Wdr8–SSX2IP complex reveals an essential link between maternal centrosome proteins and the stability of the zygotic genome for accurate vertebrate embryogenesis. Our approach provides a way of distinguishing maternal from paternal effects in early embryos and should contribute to understanding molecular defects in human infertility.

[1] Centre for Organismal Studies (COS), Heidelberg University, Im Neuenheimer Feld 230, Heidelberg 69120, Germany. [2] Zentrum für Molekulare Biologie der Universität Heidelberg (ZMBH), DKFZ-ZMBH Alliance, Im Neuenheimer Feld 282, Heidelberg D-69120, Germany. [3] Institute of Genetics, University of Bonn, Karlrobert-Kreiten-Straße 13, Bonn 53115, Germany. † Present address: Sanofi Aventis Deutschland GmbH, Industriepark Hoechst, 65926 Frankfurt, Germany. Correspondence and requests for materials should be addressed to D.I. (email: daigo.inoue@cos.uni-heidelberg.de) or to J.W. (email: jochen.wittbrodt@cos.uni-heidelberg.de) or to O.J.G. (email: ogruss@uni-bonn.de).

The animal microtubule (MT) organizing center (MTOC), or centrosome, comprises a pair of centrioles embedded in pericentriolar material (PCM)[1–4]. The centrosome duplicates during every cell cycle to define the two MTOCs of the bipolar mitotic spindle[2,3]. Most animal oocytes eliminate centrosomes during oogenesis[3]. This seems necessary to prevent their aging, parthenogenesis or abnormal embryonic mitoses that would be a consequence of an excess number of centrosomes. Upon fertilisation, the zygote forms new centrosomes from the paternal centrioles and from a pool of maternal factors that have been characterized to some extent but not been comprehensively identified[3,5,6]. Initial, accurate centrosome assembly governs the progression of early cleavages in animal zygotes. Oocytes accumulate stock-piles of maternal centrosomal factors during oogenesis, which will assemble around paternal centrioles and permit the formation of the initial zygotic centrosomes upon fertilisation[5,6]. The zygotic centrosomes serve as original templates for all the centrosomes that are created during subsequent cleavages. The establishment of the first functional centrosomes is precondition for the proper segregation of chromosomes during early cleavages and determines the entire subsequent progression of development[6].

In somatic cells in culture, bipolar mitotic spindle assembly and progression to anaphase are possible in the absence of centrosomes; under physiological conditions, however, the pair of centrosomes consistently defines mitotic spindle poles[7,8]. In contrast, the centrosomes of *Drosophila*, *Caenorhabditis elegans* and sea urchin zygotes seem to be essential for embryonic mitoses (that is, cleavages)[9–14]. The *Drosophila* PCM protein Spd-2, and the centriolar protein Sas-4, for instance, do not appear to be strictly required for somatic mitosis in cultured *Drosophila* cells and late development (after midblastula transition), but are essential for early embryonic mitoses via their centrosome assembly functions[12,13,15]. Studies in worms and flies have thus revealed the existence of regulatory complexes of centrosome proteins and their roles in the molecular steps of zygotic centrosome assembly. In contrast, the corresponding mechanisms in vertebrates remain largely obscure.

A few zebrafish mutants have served as a basis for examining the physiological functions of maternal centrosomal factors in vertebrates[16–18]. The *cellular atoll* (*cea*) embryo, a maternal-effect *Sas-6* mutant, fails to duplicate its centrosomes and has revealed functions of centrosomal factors that are conserved in human somatic cells and teleost embryos[17]. In the lymphoid-restricted membrane protein (*lrmp*) mutant *futile*, centrosomes do not attach to pronuclei immediately after fertilisation, demonstrating another maternal-specific function[18]. However, many additional centrosome proteins have only scarcely been investigated in developing vertebrates, making it impossible to model the molecular networks required for centrosome assembly during the initial stages of embryogenesis and later development. Moreover, current knockout methods have not provided a means of peeling apart the distinct functions of maternal and zygotic factors in molecular detail. Thus the role played by centrosomes in vertebrate embryogenesis, and their contributions to development and infertility mechanisms in humans, remain unclear.

To gain molecular insights into the role of maternal-specific centrosomal factors in centrosome assembly *in vivo*, we used the vertebrate model medaka (*O. latipes*)[19]. We analysed the WD40 repeat-containing protein, Wdr8, (the orthologue of human WRAP73), which we recently identified in a screen for maternal proteins[20]. We applied CRISPR–Cas9-mediated targeted gene inactivation, and elicited specific centrosome assembly defects in the absence of maternal and zygotic Wdr8. This allowed us to unravel an essential role for Wdr8 in maternal centrosome assembly. Subsequently we used mRNA injection, which mimicked maternal gene expression, and observed a remarkable reconstitution of centrosome assembly, proper cell divisions and gross development until adulthood. This *in vivo* reconstitution strategy of maternal Wdr8 functions allowed us to perform a straightforward screening of mutant variants that revealed domains/modules of the Wdr8 protein essential for centrosome assembly in living vertebrate embryos. Our system delivered molecular insights into Wdr8's essential function in embryonic centrosome assembly.

## Results

**Maternal Wdr8 is essential for proper cleavage divisions**. In a recent study we identified a number of proteins that were specifically upregulated in *Xenopus laevis* oocytes. Alongside the centrosome assembly factor SSX2IP (ref. 20), our screen revealed an upregulation of Wdr8, a previously uncharacterised WD40 repeat-containing protein, which is well conserved among species (Supplementary Fig. 1a,b). These two proteins interacted with each other in *Xenopus* egg extracts, suggesting a potentially common function[20,21].

To analyse Wdr8's functions in early vertebrate development, we took advantage of the transparency of medaka embryos to carry out a cell biological analysis combined with a CRISPR–Cas9 genome editing approach to inactivate medaka *Wdr8* (*O. latipes Wdr8*, *OlWdr8*, hereafter referred to as *Wdr8*). The inactivation was achieved through the targeted integration of a green fluorescent protein (GFP)-stop cassette into exon-3 of the *Wdr8* locus (Supplementary Fig. 2a,b). A cross of heterozygous parents yielded *Wdr8*$^{-/-}$ homozygous offspring that showed no obvious abnormalities during development and were phenotypically undistinguishable from wild-type fish from embryonic stages to adulthood (Supplementary Fig. 3). However, when we compared the four combinations resulting from crosses of *Wdr8*$^{-/-}$ and wild-type fish (Fig. 1a), we found severe phenotypic abnormalities in all zygotes from *Wdr8*$^{-/-}$ homozygous mothers but not its homozygous fathers. While the first cleavage in Wild-type (*WT*) and all crosses happened uniformly within a maximum of a 4-min time window (Supplementary Table 1), the zygotes from *Wdr8*$^{-/-}$ homozygous mothers gradually began to delay the timing of cleavages, with the variability increasing from the second cleavage cycle on. This led to cumulative delays in the 3rd and 4th cleavage cycle (Fig. 1b, Supplementary Table 1). The second division and the following cleavage cycles revealed abnormal asymmetric cleavages or frequent cytokinesis failures which were not apparent during the first division (Fig. 1a, Supplementary Figs 4 and 5). In the absence of maternally provided Wdr8, these zygotes failed to gastrulate or reach the neurula stage (St.18) (ref. 22) (Fig. 1a). In contrast, the development of all zygotes from *Wdr8*$^{-/-}$ fathers was indistinguishable from that of wild-type zygotes (Fig. 1a,b, Supplementary Table 1, Supplementary Figs 4 and 6). These results indicate that maternally provided Wdr8 is essential for faithful cleavages in the large blastomeres of the early fish embryo.

**Exogenous expression of Wdr8 can rescue *Wdr8*$^{-/-}$ zygotes**. Assuming that the presence of maternal Wdr8 during cleavage divisions governs for the progression of normal development, we next performed rescue experiments in the maternal/zygotic mutants from the cross of the homozygous mothers and fathers (hereafter called *m/zWdr8*$^{-/-}$ unless stated otherwise). *m/zWdr8*$^{-/-}$ zygotes were devoid of endogenous maternal Wdr8 during early development (Supplementary Fig. 7). We injected an mRNA encoding an enhanced yellow fluorescent

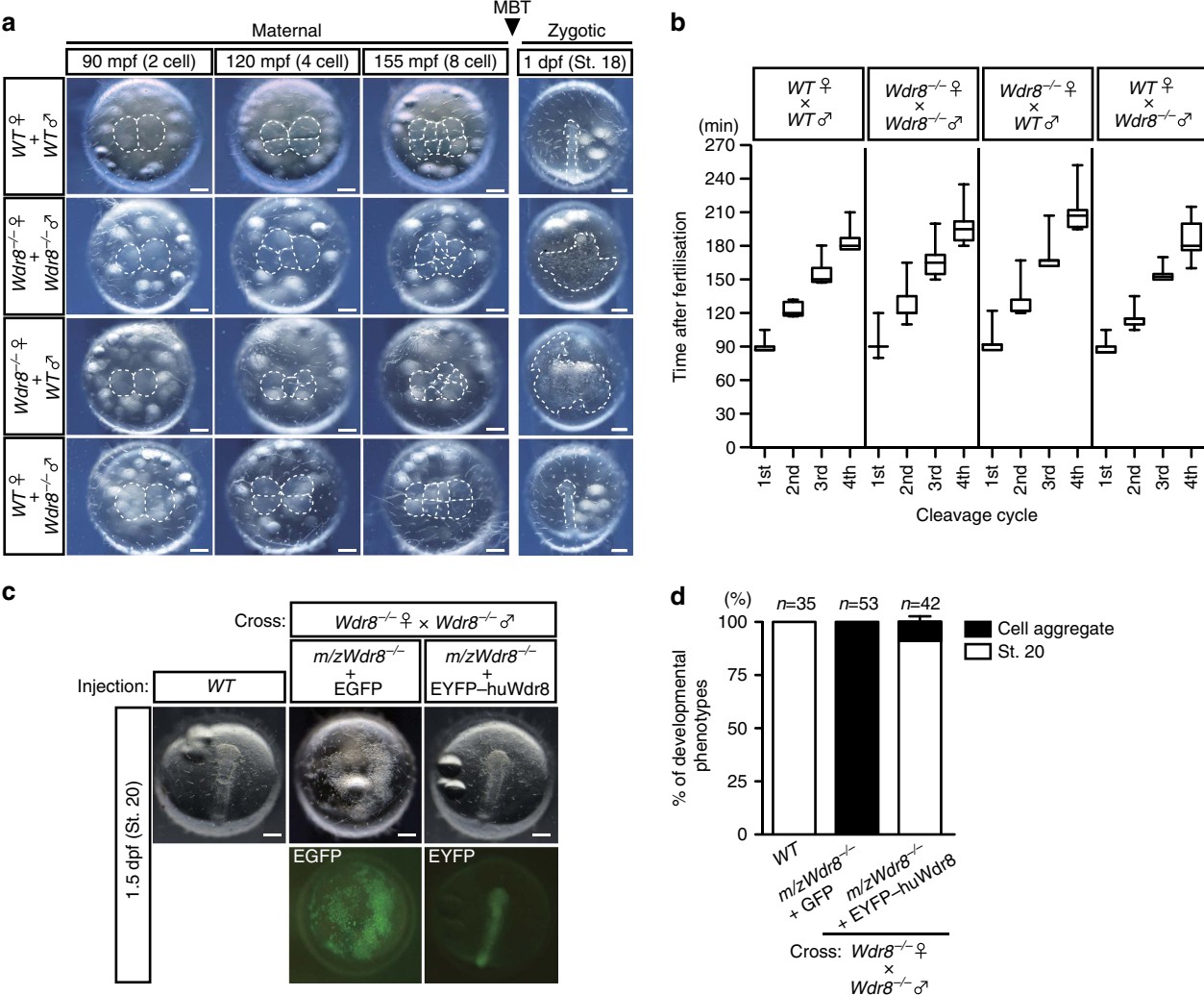

**Figure 1 | Abnormal cleavage divisions of Wdr8$^{-/-}$ zygotes are rescued by EYFP–huWdr8 expression.** (**a**) Phenotypes of paternal and maternal Wdr8$^{-/-}$ zygotes before and after midblastula transition. The maternal Wdr8$^{-/-}$ zygotes, but not the paternal, showed disordered cleavage divisions and failure of gastrulation. Dotted lines represent developing zygotes onto the yolk. (**b**) Timing of cleavage cycles in zygotes produced from the individual mating combinations in **a**. In the first cleavage cycle, there was no apparent delay between WT and the zygote from WT mother/Wdr8$^{-/-}$ father, indicating that there are no paternal contributions causing one-cycle delay[17]. On the other hand, zygotes from Wdr8$^{-/-}$ mother (crosses; Wdr8$^{-/-}$ ♀ × WT♂ and Wdr8$^{-/-}$ ♀ × Wdr8$^{-/-}$ ♂) accumulated delay and variance of the cleavage timing due to cytokinesis failure and abnormal spindle assemblies (Fig. 3b, Supplementary Fig. 4). Boxes and whiskers represent median with 25th and 75th percentile and maximum/minimum, respectively. Total zygotes and biological replicates are denoted in Supplementary Table 1. (**c,d**) Full rescue of m/zWdr8$^{-/-}$ zygotes (from Wdr8$^{-/-}$ ♀ × Wdr8$^{-/-}$ ♂) by exogenous expression of EYFP–huWdr8 (**c**) and its efficiency at 1.5 dpf (**d**). N, total embryos from three independent experiments. Data represent mean ± s.e.m. Scale bars, 200 μm. Stages are denoted as minutes/days post fertilisation (mpf, dpf) and the corresponding stages of WT are denoted in parentheses.

protein (EYFP)-fusion of the human WDR8/WRAP73 orthologue (here called EYFP–huWdr8) into m/zWdr8$^{-/-}$ zygotes within 10–15 min post fertilisation (mpf). At 70 min after injection (about 80 mpf), fluorescent proteins were detected from both the control mRNA, encoding enhanced green fluorescent protein alone, and EYFP–huWdr8. At this point zygotes underwent their first cleavage. While all the Wdr8$^{-/-}$ zygotes injected with EGFP showed abortive development, EYFP–huWdr8 expression efficiently rescued the abnormal cleavage divisions to WT-like situations, and over 90% of the injected zygotes progressed through gastrulation to subsequent developmental stages (Fig. 1c,d, Supplementary Fig. 5). The rescue was dose dependent (Supplementary Fig. 8) and required an exogenous mRNA amount about ×40 higher than that of endogenous Wdr8 mRNA to achieve a reliable rescue efficiency (Supplementary

Fig. 9). Intriguingly, the majority of the rescued embryos developed into juveniles indistinguishable from wild-type fish based on gross morphology (Discussion). These results not only confirmed the specificity of the knockout phenotype, but also demonstrated that maternal Wdr8 plays an essential role in early development by maintaining the integrity of embryonic cleavages.

**Centrosomal localisation of Wdr8 in rescued zygotes.** The fluorescent tag of the rescue construct allowed us to determine the localisation of EYFP–huWdr8 in living blastomeres. In vivo microscopy showed that EYFP–huWdr8 localised specifically to one or two distinct dot-like structures in each blastomere of the rescued zygotes (Fig. 2a,b). During the cleavage cycles from the one- to the four-cell stage, the two dot-like structures remained adjacent to each other, then

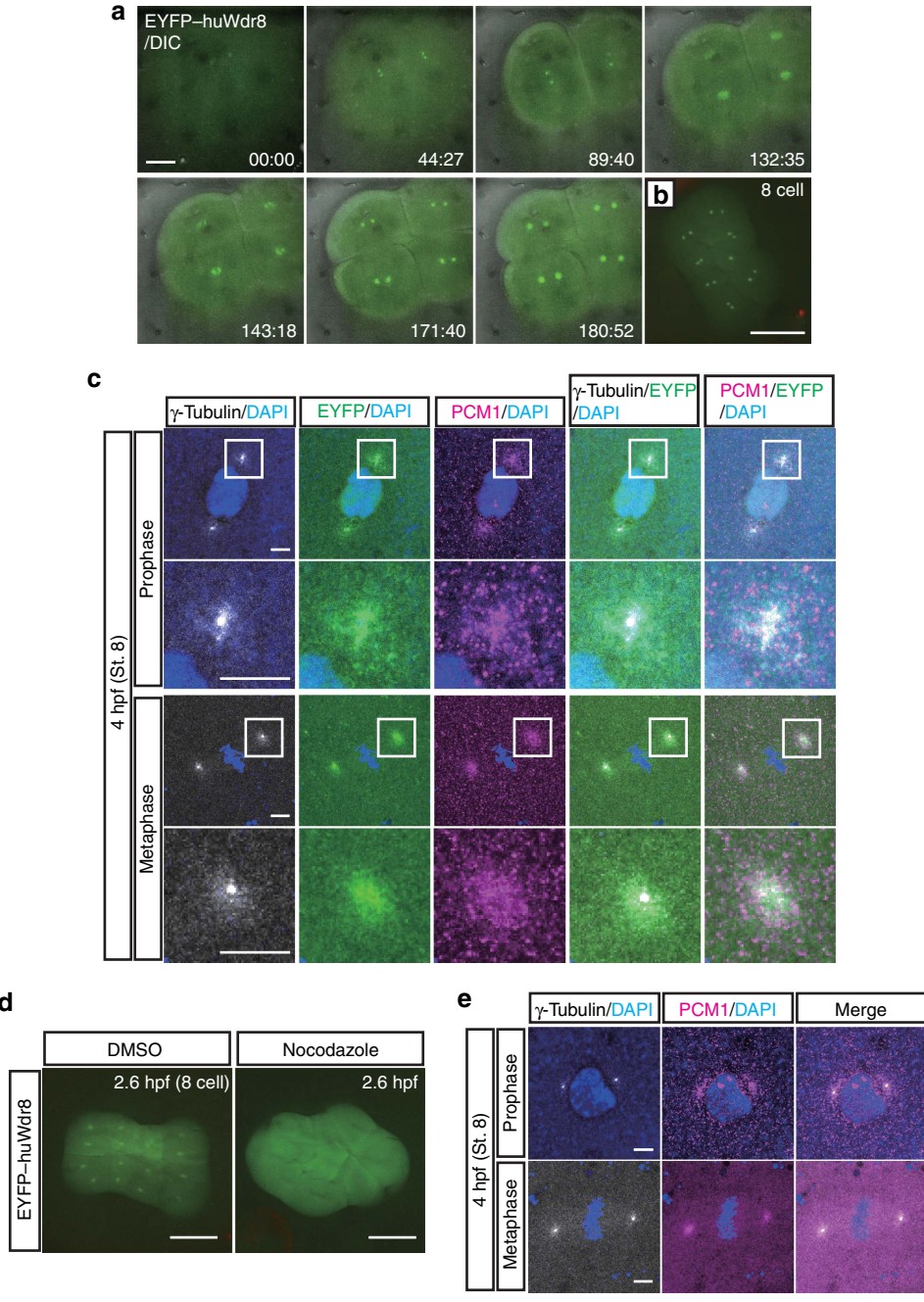

**Figure 2 | Wdr8 localises to the centrosome in the rescued *Wdr8*<sup>−/−</sup> blastomeres.** (**a,b**) Spatiotemporal localisation of EYFP–huWdr8 in the rescued *Wdr8*<sup>−/−</sup> zygotes. Time-lapse images show stereotypic centrosome cycles from one- to four-cell stage (**a**) and 8-cell stage (**b**). Time, min. (**c,e**) Wdr8 is a novel maternal centrosome protein. EYFP–huWdr8 localised next to γ-tubulin-positive PCM and co-localised with the CS protein, PCM1 (**c**, insets; magnified view of the centrosome), resembling the wild-type situation (**e**). (**d**) Centrosomal EYFP–huWdr8 in the rescued *Wdr8*<sup>−/−</sup> zygotes dispersed into cytoplasm after nocodazole treatment. Scale bars, 100 μm (**a**), 200 μm (**b,d**), 10 μm (**c,e**). Stages are denoted as hours post fertilisation (hpf) and the corresponding stages of *WT* are denoted in parentheses.

separated and migrated to opposing sides of the blastomeres during mitosis (Fig. 2a, Supplementary Movie 1). At the end of mitosis (before the emergence of the cleavage furrow), the Wdr8 dots were more dispersed (Fig. 2a, time points at 132:35 and 180:52, and Supplementary Movie 1). This localisation was strikingly reminiscent of the pattern that MTOC proteins follow during the cell cycle as they separate to form the mitotic spindle. Indeed, EYFP–huWdr8 localised next to γ-tubulin-positive PCM and co-localised with the centriolar satellite (CS)-marker PCM1

(refs 23,24) at both prophase and metaphase, demonstrating that Wdr8 is a novel maternal centrosomal and CS factor in medaka zygotes (Fig. 2c). Consistent with this, treating zygotes with nocodazole to depolymerise MTs led to the loss of Wdr8's accumulation in dot-like structures (Fig. 2d). In controls expressing EGFP alone (wild-type and *m/zWdr8*<sup>−/−</sup> blastomeres), the EGFP signal was found in the cytoplasm, confirming that the centrosomal localisation of the EYFP–huWdr8 fusion protein was due to the Wdr8 sequence (Supplementary

Movies 1–3). Even EYFP–Wdr8 generated from injection of low concentrations of mRNA ($5 \, ng \, \mu l^{-1}$) efficiently accumulated at the centrosome, suggesting that overexpression of EYFP–Wdr8 is not the reason it targets to the centrosome (Supplementary Fig. 8). Taken together, our results suggest that Wdr8 is a maternal centrosome protein required for proper embryonic mitoses.

**Wdr8 is essential for zygotic centrosome assembly**. EYFP–huWdr8 expression rescued the centrosomal localisation of γ-tubulin and PCM1 in the zygotes in a way comparable to that of wild-type zygotes (Fig. 2e). This indicated that EYFP–huWdr8 successfully rescued the phenotype of $m/zWdr8^{-/-}$ zygotes by restoring the structure of embryonic MTOCs. We therefore asked if MTOC assembly is affected in $m/zWdr8^{-/-}$ blastomeres. In these blastomeres, γ-tubulin and PCM1 were severely scattered at prophase and metaphase compared with wild type, sometimes forming multiple PCM foci that were unevenly fragmented (Fig. 3a). Furthermore, while bipolar mitotic spindles readily formed in wild-type blastomeres, the scattered PCM of $m/zWdr8^{-/-}$ blastomeres induced multipolar spindle assembly from prophase to telophase (Fig. 3b–d). Importantly, these centrosomal and spindle abnormalities led to severe chromosome alignment defects (Fig. 3a–d), which predict chromosome segregation errors and are consistent with the failures in cytokinesis that were frequently observed (Supplementary Figs 4 and 5). In contrast, blastomeres expressing exogenous EYFP–huWdr8 recovered from all three defects: abnormal centrosome assembly, multipolar spindle formation and chromosome alignment errors (Figs 2c and 3b). These results demonstrate that maternal Wdr8 ensures faithful centrosome assembly and is essential for the maintenance of functional MTOCs in embryonic mitosis.

Of note, we realised that MTOCs in the first cleavage cycle, visualised by γ-tubulin antibodies, achieved an apparently normal position at the onset of mitosis to allow proper bipolar spindle formation in both $WT$ and $m/zWdr8^{-/-}$ zygotes (Supplementary Fig. 10a,b). In the second cleavage, however, the $m/zWdr8^{-/-}$ blastomeres showed abnormally positioned MTOCs, leading to spindle assembly defects (Supplementary Fig. 10a,b). Despite the emergence of the defect in MTOCs in the second cleavage cycle, the timing of cell cycle progression in $m/zWdr8^{-/-}$ was comparable to that of wild type, indicating that the 2nd embryonic cell cycle was still synchronous within a population of animals, as was the case between $WT$ zygotes (Supplementary Fig. 10c). Taken together, these results indicate that Wdr8 is a specifically maternal centrosome assembly factor, whose functions are crucial starting with the second embryonic mitosis.

**WD40 domains are essential for Wdr8 localisation/function**. Our observation that exogenously provided Wdr8 mRNA efficiently rescued defects of $Wdr8^{-/-}$ zygotes in early cleavage stages encouraged us to carry out a structure–function analysis of Wdr8. The aim was to analyse the significance of interaction partners and the domains of Wdr8 that permit them to bind. Wdr8 is conserved across species from yeast to humans, and vertebrates exhibit four WD40 domains (Fig. 4a, Supplementary Fig. 1). WD40 domains are often involved in protein–protein interactions[25], suggesting that they might play an important role in Wdr8's function.

We selected amino acids in the WD40 domains of Wdr8 that were conserved across species and mutated both the Tryptophan (W) and the Aspartate (D) in either W196 and D197 (WD40_1) or W359 and D360 (WD40_3) to Alanine (hereafter these mutant variants are called 196/197AA and 359/360AA, Fig. 4a and Supplementary Fig. 1). To assess the functional relevance

of the WD40 domain, we performed a rescue assay by injecting mRNA into $m/zWdr8^{-/-}$ zygotes. Proteins from mRNAs encoding EYFP–huWdr8 wild-type and the WD mutant variant (EYFP-196/197AA) were expressed at similar levels (Supplementary Fig. 11). In contrast to the wild-type construct, Wdr8 lost its centrosomal localisation in both WD mutant variants and was mainly detected in the cytoplasm (Fig. 4b).

When blastomeres entered mitosis, a weak but visible centrosomal localisation of the mutant variants was observed in live imaging of EYFP-196/197AA-injected zygotes (Fig. 4c, time points at 08:11, 53:12 and 98:12, and Supplementary Movie 4). Individual blastomeres exhibited multiple foci, but they disappeared during the subsequent cleavage cycles (Fig. 4c, Supplementary Movie 4). Interestingly, in $m/zWdr8^{-/-}$ blastomeres, both WD variants were ubiquitously dispersed throughout the cytoplasm and co-localised partially with scattered γ-tubulin and PCM1 at prophase (Fig. 4d). This indicates that WD variants are hypomorphic forms of Wdr8 that inefficiently localise to the centrosome, and demonstrates that the WD40 domain structure of Wdr8 is essential for proper centrosome assembly *in vivo*.

Importantly, abnormal centrosome assembly led to multipolar spindles as well as chromosome alignment defects (Fig. 4e). This was consistent with the finding that neither of the WD mutant variants was able to rescue the early $m/zWdr8^{-/-}$ phenotypes or their subsequent abortive development (Fig. 4b). These results demonstrate that intact WD40 domains govern the centrosomal localisation of Wdr8 and are required for proper centrosome functions in early embryonic mitoses of cleavage stages.

**WD40 domains mediate Wdr8–SSX2IP complex formation**. We next took advantage of the mRNA-based *in vivo* reconstitution assay to address the molecular mechanisms underlying Wdr8's function. We hypothesised that WD40 domains serve either as centrosome-targeting domains or as modules for interactions with other proteins. We first tested whether the rescue of $m/zWdr8^{-/-}$ zygotes by WD variants was achieved by targeting them to the centrosome. We used the centrosome-targeting motif PACT to facilitate the accumulation of WD mutant variants at the centrosome[26]. Both PACT fusion constructs weakly localised to the centrosome in some zygotes (2.6 hpf in Fig. 5a). In these zygotes, however, the embryonic lethality introduced by Wdr8 knockout was still not rescued (1.5 dpf in Fig. 5a). This failure—even when Wdr8 was localised to the centrosome—strongly suggests that a threshold level of centrosomal Wdr8 is critical for Wdr8 activity, for which WD40 domains specifically mediate centrosomal targeting.

Next we tested whether WD40 domains are important for interactions with other proteins. Since Wdr8 had initially been identified as an interaction partner of the CS protein SSX2IP, we hypothesised that WD40 domains might be critical for Wdr8's interaction with SSX2IP, or other centrosomal or CS proteins. To retrieve amounts of the proteins sufficient to analyse protein-protein interactions, we carried out a biochemical analysis in *X. laevis* egg extracts. We performed Flag immunoprecipitations to analyse the proteins that co-precipitated with Wdr8 following the expression of either 2xFlag-huWdr8 wild-type (Wt), 2xFlag-196/197AA (196/197AA) or 2xFlag-359/360AA (359/360AA) by mRNA addition to cytostatic factor (CSF) egg extracts arrested at metaphase[27]. Strikingly, mass spectrometry analysis revealed that SSX2IP interacted with Wt, whereas it was unable to interact with either 196/197AA or 359/360AA (Fig. 5b). This provided further evidence that WD40 domains are essential for the Wdr8–SSX2IP interactions.

Although γ-tubulin was severely scattered in $Wdr8^{-/-}$ blastomeres, γ-tubulin and all the γ-TuRC subunits (including its regulatory protein Nedd1)[28–30] interacted equally with Wt and

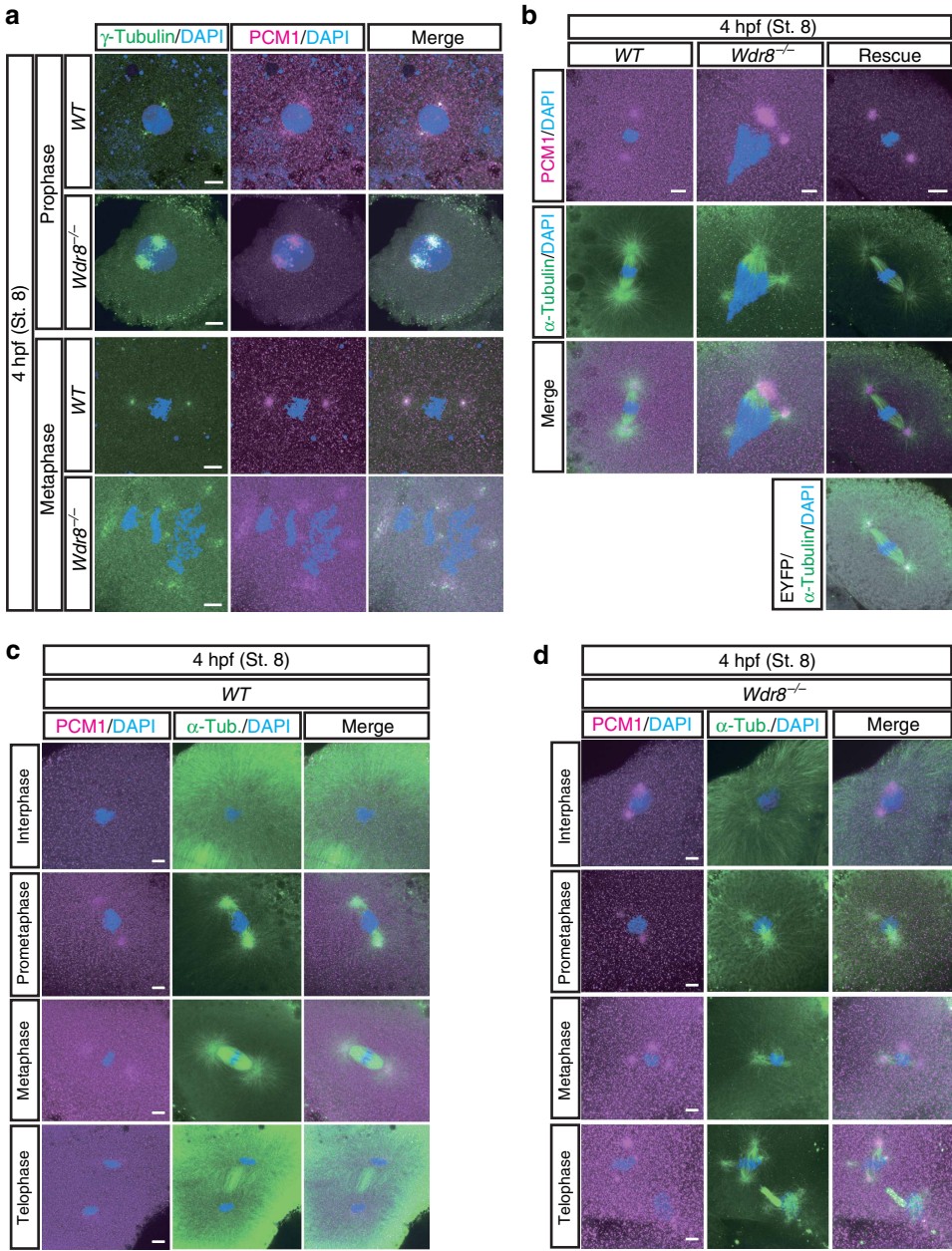

**Figure 3 | Centrosome and spindle abnormalities in Wdr8$^{-/-}$ blastomeres.** (**a,b**) Abnormal PCM and mitotic spindle assemblies in Wdr8$^{-/-}$ blastomeres. In comparison with wild-type (Fig. 2e), 100% of Wdr8$^{-/-}$ zygotes exhibited both severely scattered γ-tubulin/PCM1 (**a**) and multipolar mitotic spindles (α-tubulin) (**b**) together with chromosome alignment errors in metaphase (**a,b**). (**c,d**) Detailed process of mitotic spindle assembly in Cab and Wdr8$^{-/-}$ blastomeres. Whereas bipolar mitotic spindles were faithfully formed from prometaphase to telophase in wild-type blastomeres (**c**), fragmented centrosomes (PCM1) caused multipolar spindles (α-Tub.) as well as severe chromosome alignment defects during mitosis in Wdr8$^{-/-}$ blastomeres (**d**). These abnormalities were efficiently rescued by EYFP–huWdr8 expression (Figs 2c and 3b). Scale bars, 10 μm. Stages are denoted as hours post fertilisation (hpf) and the corresponding stage of WT are denoted in parentheses.

WD variants, indicating that WD40 domains are not required for Wdr8-γ-TuRC/Nedd1 interactions (Fig. 5b). Consistent with the mass spectrometry data, Western blot analysis confirmed that only Wt, but not WD mutant variants, interacted with SSX2IP (Fig. 5c). On the other hand, γ-tubulin's interaction with Wdr8 was independent of WD40 domains (Fig. 5c). Taken together, these results revealed that WD40 domains in Wdr8 are the basis for forming a complex with SSX2IP, but that Wdr8 interactions with γ-TuRC components are independent of the domains. In medaka blastomeres, Wdr8 wild-type almost completely co-localised with SSX2IP, strongly suggesting that Wdr8 is at

least also a CS protein which forms a complex with SSX2IP in blastomeres (Fig. 5d). In contrast, SSX2IP was remarkably dispersed in the cytoplasm in both 196/197AA- and 359/360AA-expressing blastomeres (Fig. 5d), indicating that the interaction between Wdr8 and SSX2IP is crucial for mutual localisation and the function of Wdr8–SSX2IP as a CS protein complex (Fig. 5d,e).

In summary, our results demonstrate that maternal Wdr8 is essential for centrosome assembly during the rapid mitoses in early embryonic development. It further suggests that Wdr8 carries out its functions in a complex with SSX2IP.

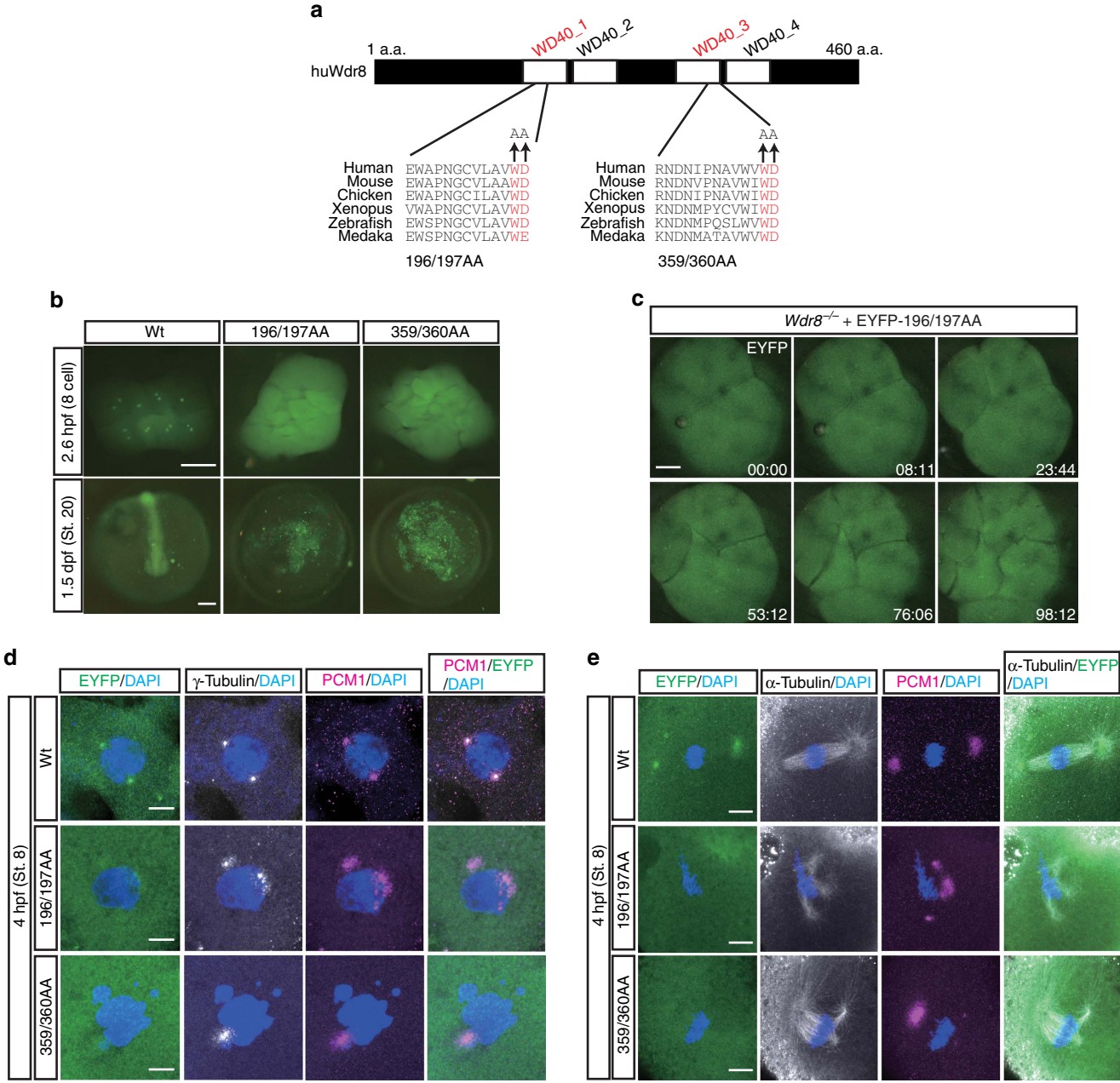

**Figure 4 | WD40 domains are essential for Wdr8 function. (a)** The conserved four WD40 domains in huWdr8. Mutations for WD mutant variants are denoted in red. **(b)** External phenotypes of WD mutant-injected $Wdr8^{-/-}$ zygotes and embryos. Note that both variants mostly localised in the cytoplasm (2.6 hpf), unable to rescue $Wdr8^{-/-}$ embryos (1.5 dpf). **(c)** Time-lapse images of EYFP-196/197AA localisation in $Wdr8^{-/-}$ blastomeres. EYFP-196/197AA weakly localised to the centrosome with multiple foci, which disappeared during cleavages. Time, min. **(d,e)**, In comparison with EYFP–huWdr8 wild-type, expression of either mutant variant was unable to rescue abnormal PCM assembly ($\gamma$-tubulin/PCM1), multipolar mitotic spindles, and chromosome alignment defects. Scale bars, 200 μm **(b)**, 100 μm **(c)** and 10 μm **(d,e)**. Stages are denoted as hours/days post fertilisation (hpf, dpf) and the corresponding stage of $WT$ are denoted in parentheses.

## Discussion

Although it has long been suggested that centrosomes are essential for embryonic mitosis in vertebrates, their ultimate impact on embryonic development and the molecular mechanisms underlying their activity have remained ill-defined. Several studies in vertebrate embryos have explored the regulation of the centrosome during embryonic mitoses[16–18,31,32]. Zebrafish maternal-effect mutants, $cea$ ($Sas$-6) and $futile$ ($Lrmp$), have revealed functions in centrosomal regulation in the vertebrate embryo[17,18]. However, due to the difficulty of efficient rescue approaches, these pioneering findings have still left challenges in determining the proteins these factors interact with and how they function.

In this study, we established a robust $in vivo$ rescue approach to reveal detailed molecular mechanisms of centrosome assembly in early vertebrates. We show that centrosome activation by maternal expression of the highly conserved protein Wdr8 (Supplementary Fig. 1a,b) is absolutely essential for mitotic spindle bipolarity and the accurate inheritance of the zygote genome. This differs from a certain dispensability of centrosomes

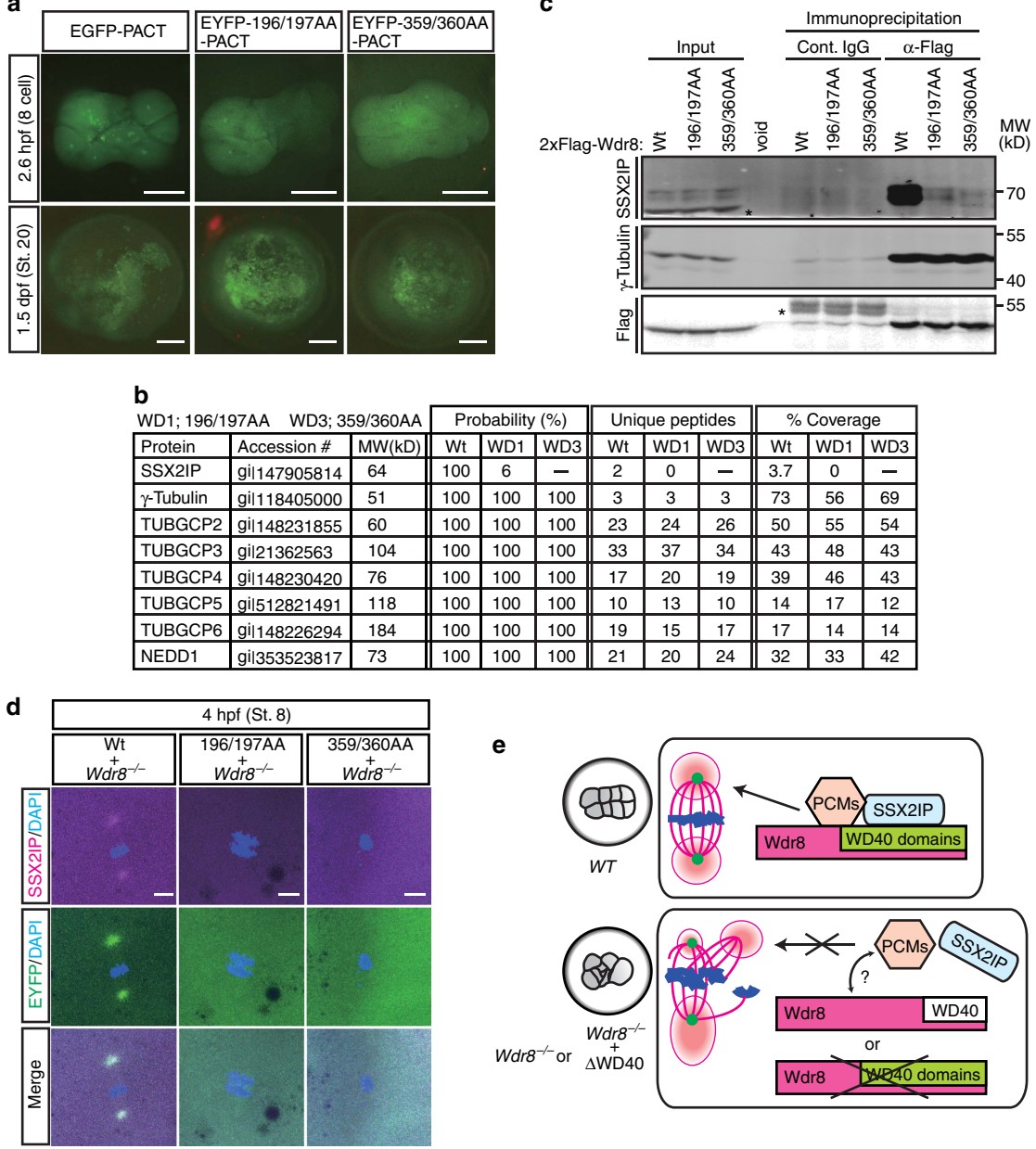

**Figure 5 | WD40 domains is essential to form Wdr8–SSX2IP complex and its localisation.** (**a**) The PACT domain could target EGFP to localise to the centrosome (positive controls) and mediated centrosome targeting of either EYFP-196/197AA-PACT or EYFP-359/360AA-PACT. However, this was insufficient for rescue of *Wdr8$^{-/-}$* embryos. (**b,c**) Identification of Wdr8 interaction factors by mass spectrometry and immunoblot analyses with *Xenopus* egg extracts. Wt, but not WD mutant variants, interacted with SSX2IP (**b,c**). By contrast, γ-TuRC subunits/Nedd1 (**b**) and γ-tubulin (**c**) interacted with Wdr8 independently of WD40 domains. *Background. (**d**) Interdependent localisation of Wdr8 and SSX2IP via WD40 domains. Due to failure of Wdr8–SSX2IP interaction, WD mutant variants, in contrast to Wt, were unable to colocalise with SSX2IP at the centrosome in *Wdr8$^{-/-}$* blastomeres. (**e**) Wdr8–SSX2IP localises to the centrosome to tether other PCM components (for example, γ-TuRC, which is essential for maternal PCM assembly). The absence of maternal Wdr8 or dysfunction of its WD40 domains causes PCM assembly defects. Scale bars, 200 μm (**a**), 20 μm (**d**). Stages are denoted as hours per days post fertilisation (hpf, dpf) and the corresponding stage of *WT* are denoted in parentheses.

in somatic cells in cell culture[7,8]. It seems likely that functional requirements for centrosomes in rapidly dividing vertebrate zygotes are far stricter than in somatic cells. While a complete loss of functions of centrosomal proteins can be compensated for by single cells in culture, multicellular cell arrangements of embryos fail to do so. This highlights the importance of exploiting *in vivo* models for the analysis of centrosomal proteins in cell division. Here, we present a first step towards the molecular analysis of the physiological significance of centrosomes in early vertebrate development.

Our CRISPR–Cas9 knockout medaka line also segregates distinct paternal and maternal mechanisms that contribute to the regulation of centrosome assembly from the first and from the second cleavage on (Supplementary Fig. 10). Successful fertilization and bipolar spindle formation in the first mitotic cell cycle of medaka *Wdr8$^{-/-}$* zygotes indicate that the sperm from *Wdr8$^{-/-}$* fathers are functional and that they are likely to carry two centrioles, which define the two spindle poles in the first mitosis. PCM assembly seems not be acutely affected in the first cell cycle of *Wdr8$^{-/-}$* zygotes, in contrast to the immediate

PCM defects upon knockdown or in mutants of the PCM components such as Spd-2 and Spd-5 (refs 33,34). The fact that mitotic defects arose in our experiments from the second cell cycle onwards independently of the genotype of the father ($Wdr8^{-/-}$ or $WT$) may confirm that sperm centrioles are still functional in $Wdr8^{-/-}$ fathers, possibly due a long-lasting maternal effect during their development. Alternatively, maternal Wdr8 may not act upstream of PCM assembly, but rather secure the integrity of PCM assembly and structure. The defect of PCM assembly upon loss of maternal Wdr8 may also be partially compensated by other, functionally similar PCM/CS proteins, which may hide the effect of Wdr8 absence in the first cell cycle but still cause the accumulation of the defect from the second cell cycle on. Although shared paternal and maternal contributions in centrosome formation seem to be conserved across species and are thus biologically important[10,14,17], the molecular details of a pathway for new centrosome assembly from paternal and maternal centrosome proteins in vertebrate embryonic mitosis remains largely unknown.

We show that the early lethality of $m/zWdr8^{-/-}$ medaka zygotes can be rescued with high efficiency either through maternal effects or the exogenous expression of functional Wdr8 mRNA, which mimics the presence of a corresponding maternal mRNA. Our studies may profit from the slower development of medaka than that of zebrafish, which may provide enough time to translate and mature exogenous proteins. Maternal mRNAs may also have specific sequences other than those of their 3′UTRs which stabilise or increase its translation efficacy[35]. Another plausible explanation for the high rescue efficiency observed here is that the requirement of Wdr8 from the second cleavage cycle on provides enough time to sufficiently accumulate the protein and to rescue $Wdr8^{-/-}$ zygotes. Strikingly, the rescued animals exhibited normal development. Our reliable 'reconstitution' rescue system, based on wild-type Wdr8 and several mutant variants, provides a robust readout that directly links the molecular features of Wdr8 structure to its function, such as its interaction with SSX2IP. This highlights the fact that reverse genetic approaches with the CRISPR–Cas9 system are well suited to dissect maternal- and zygotic-specific aspects of centrosome regulation. By tackling a key question of centrosome 'rebirth' in medaka, we introduce an approach that can be applied to many other very important questions about early oogenesis—particularly the range of processes that depend on maternal factors.

Recently, Wdr8 was shown to form a ternary complex with SSX2IP and the minus-end-directed kinesin Pkl1 (kinesin-14 homologue), which maintains minus-end pulling forces of MTs in fission yeast[36]. The SSX2IP–Wdr8–Pkl ternary complex itself is required for its localisation/function in the spindle pole body[36]. The yeast complex provides a model case in which Wdr8–SSX2IP contributes to the generation of a pulling force by the MTs at the spindle pole body, possibly by capping γ-tubulin[36,37]. There is now evidence that Wdr8 is required for cell division in human somatic cells, although no further insights about its mode of action are provided including its physiological significance in development[38]. In cell culture, the Wdr8 partner SSX2IP is shown to be required in CS for the recruitment of γ-tubulin ring complexes (γ-TuRCs) and other specific components into the PCM for the assembly of expanded mitotic PCM[20,39]. Our work here further elucidates the roles of SSX2IP as a maternal factor: maternal Wdr8 and SSX2IP interact with each other via Wdr8's WD40 domains to form a complex that seems to mediate the mutual localisation and function of the two proteins in centrosomes in medaka embryonic mitosis.

Although the mass spectrometry analysis in our experiments did not identify kinesin-14 in either SSX2IP- or Wdr8-IP

samples, the localisation of SSX2IP at centrosomes was dependent on dynein at least in *Xenopus* egg extracts[20]. Therefore, a complex of Wdr8–SSX2IP bound to dynein as a minus-end-directed motor could be a conserved molecular module whose function is crucial to ensure proper centrosome assembly and to maintain centrosome structure during embryonic mitosis. Since SSX2IP seems to be exclusive to CS[20,39], Wdr8–SSX2IP might be first recognised as a cargo by dynein at the CS granule to transport the complex, together with other PCM components, to the centrosome (Fig. 5e). The formation of the Wdr8–SSX2IP complex via WD40 domains may therefore be a critical step in centrosome assembly or the integrity of centrosome maturation (Fig. 5e)[40,41]. Intriguingly, our mass spectrometry analysis further demonstrates that Wdr8 associates with all subunits of the γ-TuRC, including its regulatory protein Nedd1, in a manner that is independent of WD40 domains (Fig. 5b). It is plausible that embryonic Wdr8 could serve, independently of SSX2IP, as a platform for γ-TuRC/Nedd1 for either its accumulation at the centrosome or in stabilising the minus-end of MT.

Defects of the assembly of the centrosome after fertilisation have been directly linked to infertility and abnormal embryonic development in humans[42,43]. So far, the causes of infertility have mainly been attributed to the abnormalities of the sperm centriole[42,43] and the effects of maternal centrosomal factors in oocytes have not yet received much attention. Considering that genomic instability in aging eggs causes infertility[44,45], it is reasonable to assume that failures in the upregulation or the activation of maternal centrosomal factors during meiosis may contribute to infertility as well. While it is well understood that misregulation of centriole factors leads to genomic instability[46–48], centrosome fragmentation is currently emerging as a cause of genomic instability as well as multipolar spindles in cultured cells[49]. Since $Wdr8^{-/-}$ zygotes showed chromosome alignment defects with multipolar spindles (Fig. 3b,d), it would be the first example in vertebrate development to provide an important link between centrosome integrity and genomic stability. However, it will be important to reveal how and when Wdr8 (−SSX2IP) coordinates other structural centriole and PCM components for centrosome integrity and how this mechanism contributes to genomic instability.

The rescue of $Wdr8^{-/-}$ embryos through either maternal effects or an introduction of exogenous Wdr8 expression causes them to undergo normal development, which suggests that zygotic Wdr8 is dispensable for later development (Fig. 1a, Supplementary Fig. 3). It is certainly still possible that zygotic Wdr8 has functions in later development beyond its role in embryonic centrosome assembly. Indeed, while recent studies demonstrated that Wdr8 is crucial for ciliogenesis in cultured cells, the question of whether Wdr8 is implicated in cilia function during development still remains to be addressed[21,50]. Alternatively, the failure of Wdr8 knockouts to produce a clear phenotype after early embryogenesis might be attributable to the presence of other, redundant factors that are expressed later.

## Methods

**Husbandry of medaka adults and zygotes.** Medaka (*O. latipes*) stocks (Cab wild-type, *WT*) were maintained according to the management of medaka breeding procedure[51]. Medaka adults, larvae and zygotes were reared at 28 °C under a 14-h light and 10-h dark cycle. Individual littermates of zygotes were incubated with 1 × embryo rearing medium (ERM) in a $\phi$6 cm petri dish by replacing the medium every 2 or 3 days. Zygotes were staged according to the Iwamatsu[22]. All fish are maintained in the closed stocks of COS at Heidelberg University. The husbandry and all the experiments were performed according to local animal welfare standards (Tierschutzgesetz §11, Abs. 1, Nr. 1, husbandry permit number 35-9185.64/BH Wittbrodt and mutagenesis permit number 35-9185.81/G-145/15) and in accordance with European Union animal welfare guidelines. The fish facility is under the supervision of the local representative of the animal welfare agency.

**Construction of materials and generation of Wdr8$^{-/-}$ medaka.** With CCTop (http://crispr.cos.uni-heidelberg.de)[52], a sgRNA was chosen to target exon-3 of *Oryzias latipes* Wdr8 (OlWdr8, NCBI reference sequence: XM_004070359.2) with least potential off-target sites in the remainder of the genome (Supplementary Table 2). Synthetic sgRNA oligos were annealed and cloned into pDR274 (addgene, #42250). Wdr8-sgRNA forward (5′-TAGGTCTCTCGAGCAGCCGGAC-3′) Wdr8-sgRNA reverse (5′-AAACGTCCGGCTGCTCGAGAGA-3′). The donor plasmid was created via Golden GATEway cloning[52,53], comprising the following sequences: (1) a specific sgRNA target site derived from GFP for *in vivo* linearisation of the donor vector (cf. sgRNA-1 in ref. 52), (2) a 665 bp homology flank corresponding to the upstream genomic sequence of the Wdr8-sgRNA target site, (3) a *GFP* variant (*GFP$^{var}$*)[52] that was silently mutated to prevent CRISPR/Cas9-mediated cleavage by the donor-linearising sgRNA[52], and (4) followed by a triple polyadenylation signal. The homology flank was cloned from Cab genomic DNA with primers: Forward (5′-GCCGGATCCCATGGTCCTCAGACTCCCTG T-3′), Reverse (5′-GCCGGTACCCGGCTGCTCGAGTGACCACACCT-3′) to facilitate cloning into the Golden GATEway entry vector, the forward and reverse primers were extended with a BamHI or KpnI restriction site, respectively. To knockout OlWdr8, *GFP$^{var}$* with stop codon was inserted by homology directed repair to generate nonsense mutation, which resulted in the C-terminal deletion of roughly 500 amino acids, that is, deletion of >75% of full-length OlWdr8 protein. One-cell stage Medaka zygotes were co-injected with 10 ng μl$^{-1}$ of donor plasmid, 15 ng μl$^{-1}$ per sgRNA (Wdr8-sgRNA and sgRNA-1 (ref. 52)), 150 ng μl$^{-1}$ Cas9 mRNA diluted in nuclease free water. Founders were screened by genotype PCR (Supplementary Fig. 2b) with FinClip protocol[54] since *GFP$^{var}$* expression could hardly be detected, possibly due to impaired protein folding of *GFP* fused with N-terminal short fragment of OlWdr8. To achieve reproducible genotype PCR results, the programme and the primer sets for the screening were fixed as below. The primers: for wild-type locus, forward (5′- AGTGTTCAAGCAGTCCAAC CA-3′) and reverse (5′-TGAGGAGACTAGTCCAATTGAGC-3′), for *GFP$^{var}$* insertion, forward (5′-AGTGTTCAAGCAGTCCAACCA-3′) and reverse (5′-GAA CTTGTGGCCGTTTACGT-3′). The genotype PCR reaction with Taq polymerase (NEB): (1) 95 °C for 1 min, (2) 95 °C for 30 s, (3) 65 °C for 30 s, (4) 72 °C for 40 s, repeated the cycle from step 2, 30 cycles. The expected sizes of PCR products for wild-type locus and *GFP$^{var}$* insertion were 1,036 and 1,101 bp, respectively (Supplementary Fig. 2a,b). Heterozygous F1 fish were crossed with Cab. Heterozygous F2 fish were crossed with each other to generate F3 homozygous fish, which developed normally and grew to viable and fertile adults. By crossing F3 homozygous parents, the maternal effect of Wdr8$^{-/-}$ homozygous zygotes (*m/z Wdr8$^{-/-}$*) (F4) was obtained to analyse the function of maternal Wdr8.

**Preparation of cDNAs, mRNAs, and sgRNAs.** EYFP–huWdr8, 2xFlag-huWdr8, and their mutant variants were cloned into pCS2$^+$ for mRNA generation. WD40 mutant variants, 196/197AA and 359/360AA, were generated by mutation PCR. To create PACT fusion construct, PACT domain (kindly provided by Elmar Schiebel, Zentrum für Molekulare Biologie der Universität Heidelberg (ZMBH), Heidelberg, Germany) was cloned and C-terminally fused immediate after a WD mutant variant or EGFP by fusion PCR. mRNAs were transcribed with SP6 mMessegae mMachine kit (Life Technologies). sgRNAs were transcribed with mMessage mMachine T7 Ultra kit (Life Technologies).

**RT-PCR and semi-quantitative PCR.** For both semi-quantitative and real-time PCR, 10 zygotes at individual time points of *WT*, Wdr8$^{-/-}$, or Wdr8$^{-/-}$ injected with 100 ng μl$^{-1}$ EYFP–huWdr8 mRNA were collected for total RNA extraction. Total RNA (500 ng) was used for random hexamer primed cDNA synthesis with RevertAid First Strand cDNA Synthesis Kit (Thermo Scientific). Semi-quantitative PCR was performed with 1 μl of cDNA (corresponding to 25 ng total RNA) and Taq polymerase (NEB). The primers used for semi-quantitative PCR were as follows: For 5′ side of OlWdr8 spanning from exon 4 to 5, forward (5′-GCAT GTCAGAAAGGTTTAGACTTCAGC-3′) and reverse (5′-CTGAGTAAATGCC AGTCGTCACAAAC-3′), for 3′ end of OlWdr8 spanning from exon 10 to 11, forward (5′-CTACTAAAAATGACAACATGGCC-3′) and reverse (5′-CTGAAA GCAACGGATGGTCGATG-3′), for RPL7 as reference gene[53], forward (5′-CGC CAGATCTTCAACGGTGTAT-3′) and reverse (5′-AGGCTCAGCAATCCTC AGCAT-3′). The PCR reaction was fixed for reproducible results: (1) 95 °C for 30 s, (2) 95 °C for 30 s, (3) 58 °C for 30 s, (4) 72 °C for 12 s, repeated the cycle from step 2, 27 cycles. In the real-time PCR reaction with SG qPCR Master Mix (2 ×) (Roboklon), 1/7 cDNA diluted from original stock (corresponding to about 3.6 ng total RNA) was used. The primers used for real-time qPCR were as below. For OlWdr8, forward (5′-CACTAAAAATGACAACATGGCC-3′) and reverse (5′-CTGAAAGCAACGGATGGTCGATG-3′), for huWdr8, forward (5′-CTTGT CTGCAAGGGAATCACCTTC-3′) and reverse (5′-CGCCAGGAGCTGCCAAT CACTGC-3′), for RPL7, the same primers as in semi-quantitative PCR. The real-time PCRs were performed with DNA Engine and OpticonMonitor3 software (Bio-Rad). PCR condition was as follows: (1) 95 °C for 10 min, (2) 95 °C for 15 s, (3) 58 °C for 30 s, (4) 72 °C for 15 s, repeated the cycle from step 2, 40 cycles. Relative quantification between OlWdr8 and huWdr8 was calculated by the comparative Ct method[55]. Scans of the original gels in semi-quantitative PCR shown in Supplementary Fig. 7 are presented in Supplementary Fig. 12a.

**Whole mount fluorescent immunostaining.** A total of 20–40 zygotes per sample were used for whole mount fluorescent immunostaining and the representative data were presented in the figures. Whole mount fluorescent immunostainings were performed with medaka blastomeres at 4 hpf (St. 8) with an universal immunostaining protocol[56], except for omission of the heating step. Briefly, the zygotes were fixed with 4% paraformaldehyde (PFA) prepared with 1 × PTw (1 × PBS at pH 7.3, 0.1% Tween) at 4 °C for 2 days. Fixed zygotes were dechorionated and equilibrated three times in 1 × PTw for 10 min each, followed by the acetone treatment at − 20 °C for 20 min. The zygotes were washed three times with 1 × PTw for 5 min each, and then incubated in blocking buffer (10% sheep serum, 0.8% Triton X-100, 1% BSA in 1 × PTw) for 1 h at room temperature. After blocking, the zygotes were incubated with appropriate primary antibodies in incubation buffer (1% sheep serum, 0.8% Triton X-100, 1% BSA in 1 × PTw) at 4 °C for 2 days and gently agitated on a turning wheel. To remove residual primary antibodies, the zygotes were sequentially washed twice with PBS-TS (10% sheep serum, 1% Triton X-100 in 1 × PBS) for 1 h each and with PBS-T (1% Triton X-100 in 1 × PBS) for 1 h. The zygotes were incubated with secondary antibodies and DAPI in the dark at 4 °C for 2 days, and then subjected to the washing steps as in the primary antibody incubation. The primary antibodies used were anti-γ-tubulin (1:200 dilution, Sigma-Aldrich, T6557), anti-PCM1 (rabbit, 1:300 dilution, a gift from A. Merdes, Université de Toulouse, Toulouse, France)[20], anti-α-tubulin (1:100 dilution, Sigma-Aldrich, T9026), anti-GFP (1:300 dilution, Thermo Fisher Scientific, A-10262), and anti-phospho-histone H3 (1:500 dilution, Millipore, 06-570) antibodies. Secondary antibodies were Alexa Fluor 488 goat anti-mouse IgG (Thermo Fisher Scientific, A11029), Alexa Fluor 488 goat anti-chicken IgY (Jackson ImmunoResearch, 703-545-155), Alexa Fluor 546 goat anti-mouse IgG (Thermo Fisher Scientific, A-11030), DyLight 549 goat anti-rabbit IgG (Jackson ImmunoResearch, 112-505-144), and Alexa Fluor 647 goat anti-rabbit IgG (Thermo Fisher Scientific, A-21245). All secondary antibodies were incubated at 1:200 dilutions. Nuclei were counterstained with DAPI (1:200 dilution from 5 mg ml$^{-1}$ stock, Sigma-Aldrich, D9564). Images were taken with the confocal microscopy (Leica TCS SPE) with either × 20 water (Leica ACS APO × 20/0.60 IMM CORR) or × 40 oil (Leica ACS APO × 40/1.15 Oil CS 0.17/E, 0.27) objectives and processed with Image J. For SSX2IP detection, medaka zygotes at one-cell stage were injected with 100 ng μl$^{-1}$ anti-xlSSX2IP rabbit antibody[20], and then subjected to immunostaining with Alexa Fluor 647 goat anti-rabbit IgG (Thermo Fisher Scientific, A-21245).

**Live imaging of medaka zygotes.** Fluorescence in injected medaka zygotes was checked at 70–90 min post injection. The zygote with chorion was incubated in 3% methyl cellulose/1 × ERM with the glass bottom dishes (MaTeck Corporation). Live imaging was performed by Leica SPE confocal microscope with 20 × water objective. Images were taken every 37–49 s (Supplementary Movies 1–4) with a 2–3 μm z-step size. Maximum z-stack projection images were processed with Image J and then converted to QuickTime movie files.

**Synchronised zygote development and the rescue experiments.** To avoid unsynchronised development from different clutches of zygotes, individual adult medaka couples were independently mated and checked for spawning. The spawning of eggs was finished <1 min after mating. The individual clutches of zygotes were collected at 5 min after spawning (that is, 5 mpf), and then independently incubated at 28 °C (except for Supplementary Fig. 10) or 30 °C (Supplementary Fig. 10). Particularly, for the rescue experiments with various mRNA constructs, the collected eggs were transferred to the precooled 0.5 × ERM medium at 5 min after spawning (or fertilization) to slow down development, and were then subjected to mRNA injection. Injection of the mRNA into one-cell stage zygotes was carried out within 10–15 min after spawning in the precooled 0.5 × ERM medium. 5–100 ng μl$^{-1}$ mRNAs encoding individual constructs were injected into Wdr8$^{-/-}$ zygotes in a volume of 1/3 of one-cell stage blastomere. After injection, the precooled 0.5 × ERM medium of the injection dish containing injected zygotes was exchanged to pre-warmed (28 °C) 0.5 × ERM (that is, at 10–15 min after spawning or 5–10 min after collection to the precooled medium), and the zygotes were incubated at 28 °C. Zygotes at 4 hpf (St. 8 in *WT*) were fixed with 4% PFA in 1 × PTw to perform whole mount fluorescent immunostainings. Images of external phenotypes of live zygotes were taken under Nikon SMZ18 binocular microscope with the NIS-Elements F4.00.00 imaging software (Nikon).

**Cleavage timing of medaka blastomeres.** Since the timing of multiple cleavage furrows in the same zygote was synchronous among blastomeres within 1 min variability (note that different zygotes showed different cleavage timings, thus causing higher variability of the timing in the collected data), we used the time point at which at least 50% of the blastomeres in a zygote showed furrow formation. For example, if six blastomeres at eight-cell stage (third cleavage) showed cleavage furrows at 153 min and the rest showed the furrows at 154 min, we indicated the time of 153 min. In turn, when a blastomere exhibited cytokinesis failure with no furrow formation (from Wdr8$^{-/-}$ ♀/WT ♂ or Wdr8$^{-/-}$ ♀/ Wdr8$^{-/-}$ ♂), we used the majority of time point that at least 50% of the rest of the blastomeres underwent cleavage furrows. For example, if one blastomere exhibited cytokinesis failure from the 3rd to 4th cleavage cycle (8-cell to 16-cell

stage in *WT*), we counted the time point of the seven other blastomeres undergoing cleavage divisions according to the above criterion.

**Cytokinesis failure.** Cytokinesis failure in the zygotes from $Wdr8^{-/-}$ mother gave rise to one and three blastomeres in the first and second cleavage divisions, respectively. The number of zygotes exhibiting cytokinesis failure was counted in each cleavage cycle. Note that even with the cytokinesis failure, those zygotes progress through the cell cycle as *WT*, and the remaining blastomeres in the zygote underwent cleavage divisions.

**Cleavage patterns at 2.6 hpf.** Cleavage patterns at eight-cell stage of *WT* (2.6 hpf) were classified into three types of cleavages with regard to axial symmetry[57]. Type-I; highly symmetric, Type-II; partially symmetric, Type-III; asymmetric with similar sizes of blastomeres. The abnormal cleavages were classified into Ab-I, 8 cells with various sizes of blastomeres, and Ab-II, 5–7 cells possibly due to cytokinesis failure or multipolar spindles. The rescued $Wdr8^{-/-}$ zygotes by expressing EYFP–huWdr8 (100 ng μl$^{-1}$) were also counted at 2.6 hpf to show the recovery of abnormal cleavages to type I–III cleavages.

**Nocodazole treatment.** $Wdr8^{-/-}$ zygotes injected with 100 ng μl$^{-1}$ EYFP–huWdr8 mRNA were treated with either 0.017 μM Nocodazole (Sigma-Aldrich) or DMSO at 120 mpf and then incubated for 30 min at 28 °C.

**Sample collection and statistics.** The sample size was not predetermined a priori due to limitation of egg production per fish. The randomisation and exclusion of the samples were not performed. The statistical test was chosen after Kolmogorov–Smirnov and *F*-tests. The significance in Supplementary Fig. 4 was calculated by Wilcoxon matched-paired signed rank test (two-tailed) and scored as follows: \*\*$P \leq 0.01$, \*$P \leq 0.05$. The analyses were performed with Prism 5 (GraphPad Software).

**Relatedness of Wdr8s.** Both phylogenetic tree and multiple alignments of Wdr8 proteins among vertebrate species were generated through ClustralW (genomeNet) and analysed with Geneious software (ver.8.0.5).

**Western blot analysis of medaka embryos at morula stage.** 400 ng μl$^{-1}$ mRNA encoding either EYFP–huWdr8 or WD mutant variant was injected into one-cell stage zygotes in a volume of 1/3 of one-cell stage blastomere. The embryos were incubated at 28 °C for 2 h, and then treated with hatching enzyme for another 2 h to remove chorion. In total, 50 embryos (around St. 8) for each were softly homogenised with 100 μl cold PBS to remove the yolk. After centrifugation at 956*g* for 2 min at 4 °C, the supernatant was removed and the pellet of embryos was frozen with liquid N$_2$ and stored at −80 °C. The pellet was lysed on ice with 25 μl RIPA buffer (50 mM Tris–HCl at pH 8.0, 150 mM NaCl, 5 mM EDTA, 15 mM MgCl$_2$, 1% Triton-X100) containing 10 μM pepstatin, 10 μg ml$^{-1}$ aprotinin, 0.1 mM PMSF, 1 mM Na$_3$VO$_4$ and 1 mM NaF. After centrifugation at 20,817*g* for 5 min at 4 °C, 20 μl of supernatant was mixed with the equal volume of 2 × Laemmli sample buffer containing 10% 2-mercaptoethanol, boiled for 10 min at 100 °C and subjected to SDS–polyacrylamide gel electrophoresis (SDS–PAGE) and Western blot analysis. An original Western blot data shown in Supplementary Fig. 11 is presented in Supplementary Fig. 12b.

**Immunoprecipitation of 2xFlag-huWdr8 from *Xenopus* CSF egg extracts.** *Xenopus* CSF egg extracts were routinely prepared from mature *Xenopus* oocytes arrested at meiotic metaphase II according to the established protocol[27]. An aliquot of 80 μl of the extracts was incubated with 3 μg mRNA of either 2xFlag-huWdr8 wild-type, 2xFlag-196/197AA, or 2xFlag-359/360AA for 90 min at 23 °C to express Flag-tagged fusion proteins in the extracts. The immunoprecipitations were performed with 2 μg of either anti-Flag antibody (Sigma-Aldrich, F1804) or normal mouse serum (Invitrogen, #1410) for 45 min at 23 °C, followed by incubation with 20 μl of Protein G Sepharose 4 Fast Flow (GE Healthcare Life Sciences) slurry (in CSF-XB) for another 25 min at 23 °C by tapping every 5 min. The Protein G Sepharoses were washed twice and one time with 250 μl of TBS-T (10 mM Tris–HCl at pH7.5, 0.5% tween-20, 150 mM NaCl) and 250 μl of TBS (10 mM Tris–HCl at pH7.5, 150 mM NaCl), respectively. After removing TBS, the samples were prepared with 30 μl of 2 × Laemmli sample buffer containing 10% 2-mercaptoethanol, boiled for 10 min at 100 °C, subjected to Western blot and mass spectrometry analyses. Scans of the original Western blots shown in Fig. 5c are presented in Supplementary Fig. 12c.

**Dimethyl labelling and mass spectrometry.** For quantitative mass spectrometry analysis, comparison of control IgG pull-down and 2xFlag-huWdr8 pull-down samples were performed by after dimethyl labelling using stable isotopes[58,59]. Original mass spectrometry data were analysed using Proteome Discoverer 1.4 and Mascot (Matrix Science; version 2.4)[20,58]. Identified proteins were displayed by

Scaffold_4.4.3 (Proteome Software Inc.) to retrieve significantly interacted proteins with either Wdr8 wild-type or WD40 mutant variants.

**Data availability.** The data that support the findings of this study are available from the corresponding authors upon reasonable request.

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

## Acknowledgements

We thank Elmar Schiebel and Peng Liu for providing constructs and *Xenopus* egg extracts. We also appreciate Gislene Pierre and Wenbo Wang for providing constructs and sharing unpublished results. We thank Russ Hodge for critical reading and input of the manuscript. D.I. was supported by the Human Frontier Science Program (HFSP) long-term fellowship. O.J.G. was recipient of a start-professorship of the German excellence initiative, ZUK 49/TP1-16, as part of ZUK 49: Institutional strategy to promote top level research awarded to Heidelberg University. This project was supported by the ERC Advanced Grant—Manipulating and Imaging Stem Cells at Work (J.W.).

## Author contributions

O.J.G. and J.W. supervised the project. D.I and O.J.G. conceived and designed the project with critical input from J.W. D.I. performed all the experiments and analysed the data except for mass spectrometry analysis. M.S. and T.T. established CRISPR–Cas9 system. T.R. performed mass spectrometry analysis. F.B. identified Wdr8 from *Xenopus* egg extracts. D.I. and O.J.G. wrote the manuscript. J.W. supervised and supported D.I.

## Additional information

**Competing financial interests:** The authors declare no competing financial interests.

