## [Peer Review File · Nature Communications]

Reviewers' Comments:

Reviewer #1 (Remarks to the Author)

In this manuscript, the authors document that maternal expression of Wdr8 is required for vertebrate embryonic mitoses. They characterized Wdr8 as a novel maternally centrosomal protein essential for PCM assembly during embryonic mitoses and for early embryogenesis in medaka by CRISPR-Cas9. The most interesting observation in this study, the authors found the exogenous expression of wild-type Wdr8 could fully rescue Wdr8^{-/-} embryos to adulthood. Taking this opportunity, the authors found the WD40 domains in Wdr8 are critical for its function in PCM assembly and the interaction of SSX2IP by expressed the WD40 domain mutants in medaka embryos. In general, the evidence provided in this study is sufficient to support the author's conclusion and these results are potentially interesting. The manuscript suffers from numerous problems in the presentation of data. There are several major issues that have to be addressed before the manuscript can be considered for publication in Nature Communications.

Major issues:

Comment #1:

There are gross exaggerations and inaccurate statements in numerous sentences in the manuscript. The authors frequently used "clearly" in the manuscript, however, in my opinions, there are still many issues should be figured out when the authors used "clearly" to describe their observations. Thus, the authors should be careful to use this word only if the authors can provide all of evidences to support their statements. Also, the authors should be cautious to use other adjective and adverb, such as "just" in line 130, "functional" in line 131, "fully" in line 134. The authors only observed the the localisation of Wdr8 centrosomal in rescued Wdr8^{-/-} zygotes and concluded "Wdr8 acts as a centrosomal organiser in embryonic mitoses" in line 159. This statement is overstated. The over statement is also in the paragraph at last of page 11.

Comment #2:

The expression of Wdr8 in the normal and mutant should be provided during Medaka embryonic development.

Comment #3:

The authors stated "maternal Wdr8 is essential for symmetric cleavages of Medaka embryos". However, the data for supporting the symmetric cleavages is very limited and not clear (Fig.1a). Additional experiments should be performed to support the symmetric cleavage of the embryos.

Comment #4:

The authors should present the localization of Wdr8 in normal Medaka embryos in Fig. 2. The quality of Fig. 2 b and c is very low, and these data should be replaced with high quality pictures and the stage of the embryos was not clear.

Comment #5:

The authors should provide the control results and clearly describe the stage of the embryos in Fig. 3. Fig. S4 should be integrated into Fig. 3. Additionally, severe chromosome instability was not observed in Wdr8^{-/-} blastomeres from Fig. S4b.

Comment #6:

The stages of WD mutant-injected Wdr8^{-/-} embryos were not clear and should be addressed by other approaches in Fig. 4a. The controls should be provided in Fig. 4d and e, and the stage of the embryos should be described.

Comment #7:

"Both PACT fusion constructs weakly localised to the centrosome in some embryos" seems unusual. Thus, a positive control of PACT fusion constructs to the centrosome should be provided in Fig. 5a.

Comment #8:

Although the authors found that the WD40 domains of Wdr8 were critical for its function and the interaction between Wdr8 and SSX2IP. However, there are still lack of solid evidences to support the conclusion-"Wdr8-SSX2IP complex carries maternally essential functions for PCM assembly to ensure rapid embryonic mitoses". The injection of PACT fusion constructs of SSX2IP to Wdr8-/- embryos is encouraged to address this issue.

Comment #9:

The authors did not provide any evidence to address the questions of human infertility in this study. Thus, "Findings and methodology shown here provide a route to ultimately addressing questions concerning the causes of human infertility" in the abstract should be avoided.

Other minor issues: "OIWdr8" in line 106 is not correct. The meaning of "the two domains" in line 147 is not clear. What is the meaning of "359/360AA" in line 193. Some words and characters was lost in lines 247, 315, 426.

Reviewer #3 (Remarks to the Author)

Review of manuscript NCOMMS-16-01525 by Inoue et al.

The authors investigated the function of the WD40 repeat containing protein Wdr8 by inactivating the gene using CRISPR/Cas9 in medaka. The progeny of Wdr8-/- mothers exhibited delayed and abnormal cleavage divisions, often with multipolar mitoses. These phenotypic traits were rescued by injecting mRNA encoding a fusion between EYFP and human WDR8. EYFP-hWDR8 localized to one or two dots in each blastomere that coincided with the focus of gamma-tubulin and PCM1, indicating that EYFP-hWDR8 is enriched at centrosomes. The authors also found that the distribution of gamma-tubulin and of PCM1 was scattered in the progeny of Wdr8-/- mothers, instead of being present in a focus as in the wild-type. These observations led the authors to conclude that Wdr8 is required for the maintenance of functional MTOCs during embryonic mitoses. Additional experiments revealed that EYFP-tagged WD40 repeat domain mutants do not localize to centrosomes and also do not sustain phenotypic rescue amongst the progeny of Wdr8-/- homozygous mothers. The authors then tested whether the WD40 domains of Wdr8-/- direct the protein to centrosomes. To this end, they fused the centrosomal localization signal PACT to Wdr8 WD40 mutants; however, this did not yield rescue. The authors then tested whether the Wdr8 WD40 domains may be important through an association with the centrosomal partner protein SSX2IP, with which it interacts in other stems. Compatible with this view, the authors conducted co-immunoprecipitation experiments, which revealed that SSX2IP interacts with wild-type Wdr8, but not the WD40 domains mutant variants. Moreover, the authors found that Wdr8 colocalized with SSX2IP in the embryo, and that the two proteins depended on another for proper centrosomal localization.

This is an interesting study that sheds light on the mechanisms through which centrosomes assemble in the early vertebrate embryo. The work is well executed for the most part. Despite the fact that some related information was known previously (e.g. about the interaction between Wdr8 and SSX2IP in other systems), publication is recommended provided the authors address the following outstanding issues in full.

Main points:

1) Why is there a delay in cell cycle progression, in particular at first cleavage, in zygotes derived

from Wdr8^{-/-} mothers? Do the authors think that this is due merely to a longer time spent in mitosis? If so, can this be bypassed by inactivating the Spindle Assembly Checkpoint? An alternative possibility is that Wdr8^{-/-} exerts centrosome-independent functions as well.

2) The authors should consider raising antibodies against medaka Wdr8. Perhaps expression of EYFP tagged HuWdr8 does not faithfully mimic that of the endogenous protein (despite the observed rescue), in particular if expression levels differ and/or if the presence of the tag perturbs localization. For instance, it may be that the centrosomal enrichment is observed only upon overexpression of the protein.

3) Related to the above comment, how does the amount of injected EYFP-HuWdr8 compare with those of endogenous Wdr8? At the minimum quantitative RT-PCR experiments should be conducted to clarify this point.

4) Besides testing the PACT domain as a means to target WD40 mutant variants to the centrosome, the authors should consider testing also the centrosomal localization signals from cyclin E (CLS; PMID: 15514162) and that encoded by exon 9 of human Sgo1 (CTS; see PMID: 26365192). Fusion to the PACT sequence with the WD40 mutant variants resulted in only weak centrosomal localizatio, but perhaps stronger localization to centrosomes achieved by the CLS or the CTS could lead to more substantial rescue.

5) The authors assert that EYFP-HuWdr8 and SSX2IP co-localize, supporting the notion that they are part of a complex. Conventional confocal microscopy is not adequate to determine this -a super-resolution approach would be needed to reach this conclusion with certainty.

6) The authors propose that the WD40 domains of Wdr8 mediate complex formation with SSX2IP, and thus drag the latter protein to centrosomes. If this is true, then fusing the WD40 domain mutants to SSX2IP should rescue the phenotype incurred following loss of Wdr8 function. Has this experiment been attempted?

7) In many place, the number of embryos examined is missing; stating "all embryos" (e.g. on l. 113) without stating the number of embryos analyzed is insufficient. Likewise, numbers must be given to qualify statements such as "... failure of cytokinesis... which was sometimes observed".

Smaller points:

8) Since the authors use human Wdr8 to test function in medaka, and then the human protein to test interactions in *Xenopus* extracts, they should provide a Supplementary Figure to show the relatedness of these proteins (i.e. Wdr8 and SSX2IP proteins).

9) The authors should test whether the localization of EYFP-HuWdr8 around centrosomes is dependent on intact microtubules.

10) It is puzzling that Wdr8 should not be required for PCM assembly in somatic cells, because such a requirement also exists in human somatic cells (reference 34 quoted in the manuscript under consideration here). The authors might want to comment a bit more on this point.

11) On numerous occasions, the authors state that their findings "clearly demonstrate" this or that. It appears that writing "demonstrate" is more than enough.

Minor points/typos:

- l. 155-159: awkward sentence; please clarify/polish
- l. 181-183: awkward sentence; please clarify/polish

- l. 272-274: somewhat strange concluding sentence; please clarify/polish
- l. 448: incomplete sentence
- In the legend of Fig. 1A, please spell out what the stippled areas represent.
- Does Fig.1b truly report minima and maxima?
- l. 651, "by contrast" is meant instead of "contrary".

Reviewer #4 (Remarks to the Author)

Inoue et al characterize the function of an identified maternally contributed centrosomal protein, Wdr8, in the early medaka embryo. They generate a CRISPR/Cas9 allele in this gene and find that absence of maternal Wdr8 function leads to defects in cell division, such as aberrant and delayed cleavage followed by aborted embryogenesis. They also begin to characterize the interactions between Wdr8 and other proteins involved in centrosomal assembly, finding that Wdr8 binds SSX2IP as well as gamma tubulin and PCM1. They are able to express functioning Wdr8 protein in the early embryo and, using this assay and in vitro mutagenized proteins, address the role of the WD40 domains in Wdr8, and find that the SSX2IP interaction is dependent on the integrity of this domain yet that with gamma-tubulin and PCM1 is not. Together with localization of the endogenous and expressed proteins, the manuscript presents a model for the localization of these proteins in centrosomes as they function in centrosome assembly.

Major

- The presentation of the parental vs zygotic effects of Wdr8 loss of function does not clearly state whether there is a paternal contribution (independent of a maternal one), which may lead to a one-cell cycle delay in the first cell cycle, as shown in *C. elegans* and zebrafish for mutations in centriolar components. Images of embryos from mutant fathers are shown in Fig 1a but there is no timeline so that it is not possible to assess this. Such a delay would result in tetraploid embryos. Embryos from Wdr8 mutant fathers are shown to be normal in Supp Fig. 2, so presumably there is no delay (because tetraploid embryos may not be viable), however, the timing of the cell division is the primary assay in the manuscript and this is not clearly shown. Given the behavior of centriolar genes, specifically addressing this would be important. It would be interesting to compare to the behavior of paternal mutations in centriolar genes (delay in first cell cycle) and centrosomal genes (presumably no delay, as reported in this manuscript).
- Why do Wdr8 maternal mutants exhibit defects in cell cleavage (furrow positioning?) This phenotype is not addressed in the manuscript. Models of cell cleavage positioning have been proposed (Wühr et al, 2010) and such models may help explain the observed phenotype.
- The proposed delay in cell cleavage cycle in Wdr8 maternal mutant embryos contrasts with the general behavior of centriolar defects, in which cell cycles appear to occur synchronously, but are skipped if the cell cycle is defective. In particular Figure 1b, which compares the timing of cleavage in wild type and maternal Wdr8 mutant embryos shows: a) a delay in the cycling and b) an increase in variance. One concern is that the delay in cycling is a consequence of a possible arrest of the first cell cycle if the fathers are mutant (which is not possible to discern from the figure legend), so that the first cell division in mutants occurs at the time of the first cell division in wild type and so on. Another concern is that the increase in variance could be caused by having a smaller pool of mutant embryos to study, which may have artefactually led to more asynchronous data. The manuscript methods do not clearly state how embryos were synchronized so that this is also difficult to assess (the subsection of the methods that addresses Rescue states that "eggs were immediately collected after checking mating" - one concern is that if homozygous females lay eggs more irregularly that may increase cell division variance. IVF would be the more safe-proof way to insure synchronicity. It is also not clear if the data IN Fig. 1B is from embryos which are individually followed through time (which could independently confirm that indeed there is an increase in variance in cell cycle duration) or as a group (where variance could be due to initial asynchronicity in fertilization).
- Related to this, why is there no variance for the 1st and 4th divisions for the mutant in Figure 1b?

- Fusions to the PACT centrosome targeting domain are used to localize the WD mutants proteins to the centrosome. The lethality is not rescued and the authors show that localization to centrosomes is only weak. The fact that the proteins are (to some extent) localized to the centrosomes leads the authors to conclude that the WD40 domains are not required for only centrosome targeting. However, function could require a threshold level of localization. In addition, the authors subsequently show that WD40 domains interact with SSX2IP - could it be that that interaction is mediating Wdr8 centrosome localization? Perhaps the wording could be modified here so that the logic does not appear to have these caveats.

- The Discussion states that rescued Wdr8 animals grow to be fertile adults. The fact that these adults are fertile is important for the reasons outlined above (on potential cell division lags and ploidy duplication) and the information should be presented in the Results and not mentioned in the Discussion without having previously presented data.

- Number of embryos tested is not clearly specified in all experiments (text or figures)

Minor

Line 45: eliminating centrosomes in oocytes seems necessary to prevent "abnormal embryonic mitosis that would produce excess number of centrosomes". The logic here seems to be reversed, as elimination of maternal centrioles are important to prevent excess centrosomes in the early embryo, which would lead to abnormal mitoses

line 78: typo "embryogenesis"

line 306: "the localization of SSX2IP".... localization to centrosomes?

The manuscript wording at times appears to generate overstatements and would be improved if toned down

lines 46-47: "a pool of so far unidentified maternal factors" (many factors are known in various model systems involved in centriolar and centrosome assembly in the embryo)

lines 264-267: this statement as written seems to undermine the value of forward genetic efforts, and clearly forward and reverse approaches have different strengths and weaknesses. Perhaps a more even-handed way of presenting the advantage of the author's approach is the fact that after targeting a specific gene, the authors can recapitulate the rescue of the event through RNA injection immediately after fertilization. This ability in medaka does provide an undoubted advantage compared to other systems where injection of exogenous RNA may not be effective due to a time delay of translation from injected mRNA. It would be interesting if the authors could comment on this difference and why they think such a fast rescue is possible in medaka (for example, does this approach work in medaka because they are slower dividing cells?). Would this ability to rescue be expected to be a general phenomenon of medaka (as suggested in lines 288-289) or a peculiarity of Wdr8?

Lines 268-269: "we for the first time reveal the detailed molecular mechanism of centrosome assembly". This should be toned down, considering the multitude of factors involved in centrosome assembly and that the manuscript addresses the function of a small subset of these factors.

Previous studies in various model systems have shown chromosome segregation errors when centrioles are misregulated and it would be appropriate if the manuscripts discusses such studies and relates findings (and novelty of) to those prior studies.

Response to referees (NCOMMS-16-01525A)

Reviewers' comments:

Reviewer #1 (Remarks to the Author):

In this manuscript, the authors document that maternal expression of Wdr8 is required for vertebrate embryonic mitoses. They characterized Wdr8 as a novel maternally centrosomal protein essential for PCM assembly during embryonic mitoses and for early embryogenesis in medaka by CRISPR-Cas9. The most interesting observation in this study, the authors found the exogenous expression of wild-type Wdr8 could fully rescue Wdr8^{-/-} embryos to adulthood. Taking this opportunity, the authors found the WD40 domains in Wdr8 are critical for its function in PCM assembly and the interaction of SSX2IP by expressed the WD40 domain mutants in medaka embryos. In general, the evidence provided in this study is sufficient to support the author's conclusion and these results are potentially interesting. The manuscript suffers from numerous problems in the presentation of data. There are several major issues that have to be addressed before the manuscript can be considered for publication in Nature Communications.

Major issues:

Comment #1:

There are gross exaggerations and inaccurate statements in numerous sentences in the manuscript. The authors frequently used "clearly" in the manuscript, however, in my opinions, there are still many issues should be figured out when the authors used "clearly" to describe their observations. Thus, the authors should be careful to use this word only if the authors can provide all of evidences to support their statements. Also, the authors should be cautious to use other adjective and adverb, such as "just" in line 130, "functional" in line 131, "fully" in line 134. The authors only observed the the localisation of Wdr8 centrosomal in rescued Wdr8^{-/-} zygotes and concluded "Wdr8 acts as a centrosomal organiser in embryonic mitoses" in line 159. This statement is overstated. The over statement is also in the paragraph at last of page 11.

We appreciate this comment to improve the presentation of our results with precise statements. As the reviewer suggested, we omitted the adverb "clearly" and corrected the other statements point by point to avoid the impression of over-interpreting the data.

line 130: now line 139, corrected
line 131: now line 140, corrected
line 159: now line 176, corrected
The last paragraph of page 11: now page 13, lines 299-311, corrected

Comment #2:

The expression of Wdr8 in the normal and mutant should be provided during Medaka embryonic development.

We have now performed semi-quantitative qPCR to show the expression pattern of maternal OIWdr8 during development in both WT and m/zWdr8^{-/-} zygotes, which is now presented in Supplementary Fig.7. In order to address the depletion of maternal Wdr8 mRNA after CRISPR-Cas9 knock-out, we used the primer sets close to the 5' and the 3' ends of OIWdr8 transcript (see details, Supplementary Fig.7 and the materials and methods). In all stages analysed, full-length OIWdr8 mRNA was non-detectable in Wdr8^{-/-} zygotes, while readily visible in wt, indicating that there is no maternal Wdr8 protein expressed in Wdr8^{-/-} zygotes.

Comment #3:

The authors stated "maternal Wdr8 is essential for symmetric cleavages of Medaka embryos". However, the data for supporting the symmetric cleavages is very limited and not clear (Fig.1a). Additional experiments should be performed to support the symmetric cleavage of the embryos.

This is a valid comment. We re-evaluated and extended the data set to better describe our observations about cleavage defects in Wdr8^{-/-} fish. The new data set is now displayed in Supplementary Fig. 5. Based on a recent publication (Kraeussling, M., Wagner, T. U. & Schartl, M., *PLoS One*, 2011) we classified the patterns of cleavages at 8-cell stage (2hpf) into three types: I; highly symmetric, II; partially symmetric, III; asymmetric. In turn, we classified abnormal cleavages in Wdr8^{-/-} into two types: Ab-I; 8 cells with irregular sizes of blastomeres, Ab-II; 5-7 instead of 8 cells. By using these criteria, our new results show that 97% of Wdr8^{-/-} zygotes exhibited abnormal cleavages with various numbers and sizes of blastomeres, possibly due to cytokinesis failure and multipolar spindle assembly. These abnormal cleavages were efficiently recovered by expression of exogenous Wdr8.

Comment #4:

The authors should present the localization of Wdr8 in normal Medaka embryos in Fig. 2. The quality of Fig. 2 b and c is very low, and these data should be replaced with high quality pictures and the

stage of the embryos was not clear.

We have replaced all the panels with revised images, although our original figures submitted did not show major problems. In the revised version of the manuscript, the stages of the embryos shown here and in other panels are now clearly indicated. Whenever morphology based developmental stages were inconclusive (i.e. in phenotypically affected mutants), we have used time after fertilisation (hours post fertilisation, hpf) to allow comparison.

Comment #5:

The authors should provide the control results and clearly describe the stage of the embryos in Fig. 3. Fig. S4 should be integrated into Fig. 3. Additionally, severe chromosome instability was not observed in *Wdr8*^{-/-} blastomeres from Fig. S4b.

As suggested, we have revised Fig. 3: the revised version now contains the indication of stages along with the data that was initially presented in Supplementary Fig.4. We have also revised the wording when describing the data presented and now state the following: "...Importantly, these centrosomal and spindle abnormalities led to severe chromosome alignment defects (Fig. 3a, b, c, and d), which predicts chromosome segregation errors and is consistent with the frequently observed failures in cytokinesis" (p.8, lines 188-191).

Comment #6:

The stages of WD mutant-injected *Wdr8*^{-/-} embryos were not clear and should be addressed by other approaches in Fig. 4a. The controls should be provided in Fig. 4d and e, and the stage of the embryos should be described.

As in experiments concerning Fig. 2 and 3, we have indicated the embryonic stages in Fig. 4 using the time after fertilisation (hpf; hours post fertilisation, dpf; days post fertilisation). Separate controls have been added to complement Fig. 4d and e.

Comment #7:

"Both PACT fusion constructs weakly localised to the centrosome in some embryos" seems unusual. Thus, a positive control of PACT fusion constructs to the centrosome should be provided in Fig. 5a.

We have used PACT-GFP as a positive control for the efficiency of centrosomal accumulation via this motif. Respective figure panels have been added to Fig. 5a.

Comment #8:

Although the authors found that the WD40 domains of Wdr8 were critical for its function and the interaction between Wdr8 and SSX2IP. However, there are still lack of solid evidences to support the conclusion-"Wdr8-SSX2IP complex carries maternally essential functions for PCM assembly to ensure rapid embryonic mitoses". The injection of PACT fusion constructs of SSX2IP to Wdr8^{-/-} embryos is encouraged to address this issue.

This is a very good suggestion, which we have followed and injected an SSX2IP version fused to PACT or other centrosomal targeting signals (centrosomal localisation signal of cyclin E1 (CLS) and centrosomal targeting signal (CTS) of Sgo1). However, SSX2IP alone and the fusion proteins localised to the centrosome weaker than EYFP-huWdr8 and neither of these constructs could rescue the Wdr8^{-/-} phenotype (please see attached data below). Based on these results, we hypothesise that Wdr8 is required not only as a receptor to efficiently localise SSX2IP to the centrosome, but needs to be present in the complex even beyond the centrosomal targeting. We still believe that variants of Wdr8, which neither interacted with SSX2IP nor rescued lethality in Wdr8^{-/-} zygotes support our idea of an important function of the SSX2IP-Wdr8 complex in early embryogenesis. Nevertheless, we have toned down our statement and now phrase the following: "In summary, our results demonstrate that maternal Wdr8 is essential for PCM assembly during the rapid mitoses in early embryonic development. It further suggests that Wdr8 conducts its functions in a complex with SSX2IP" (p.12, lines 286-288).

Data 1 | Centrosomal targeting of SSX2IP with various centrosomal targeting motifs to rescue $Wdr8^{-/-}$ zygotes. External phenotypes (**a**) and rescue efficiency (**b**) of $Wdr8^{-/-}$ zygotes injected with various EGFP-SSX2IP constructs. All the EGFP-SSX2IP constructs localised to the centrosome considerably weaker than EYFP-huWdr8, independently of the centrosomal targeting motifs (2.6hpf, 8 cell in the rescue). None of the EGFP-SSX2IP constructs was able to rescue $Wdr8^{-/-}$ zygotes at 1.5dpf. Scale bars, 200 μ m. n, total number of zygotes from three independent experiments.

Comment #9:

The authors did not provide any evidence to address the questions of human infertility in this study. Thus, "Findings and methodology shown here provide a route to ultimately addressing questions concerning the causes of human infertility" in the abstract should be avoided.

We have toned down this statement as follows: "Findings and methodology shown here promise to contribute to our understanding of the molecular defects underlying human infertility" (p.2, lines 36-38).

Other minor issues:

"OIWdr8" in line 106 is not correct.

We explain now in line 107: *Oryzias latipes* Wdr8, OIWdr8

The meaning of "the two domains" in line 147 is not clear

We alternatively state "dot-like structures" in line 157.

What is the meaning of "359/360AA" in line 193

That was a typo in line 193, now "359/360AA" is deleted in line 224.

Some words and characters was lost in lines 247

New text: "...are the basis for forming a complex with SSX2IP, but that Wdr8 interactions with γ -TuRC components are independent of the domains." (p.12, lines 278-279).

315: New text: "...Wdr8 associates with all subunits of the γ -TuRC, including its regulatory protein Nedd1, in a manner that is independent of WD40 domains (Fig. 5b)" (p.15/16, lines 373-375).

426: We change "Quick Time" to "QuickTime" in line 518.

Reviewer #2 (Remarks to the Author):

Review of manuscript NCOMMS-16-01525 by Inoue et al.

The authors investigated the function of the WD40 repeat containing protein Wdr8 by inactivating the gene using CRISPR/Cas9 in medaka. The progeny of Wdr8^{-/-} mothers exhibited delayed and abnormal cleavage divisions, often with multipolar mitoses. These phenotypic traits were rescued by injecting mRNA encoding a fusion between EYFP and human WDR8. EYFP-hWDR8 localized to one or two dots in each blastomere that coincided with the focus of gamma-tubulin and PCM1, indicating that EYFP-hWDR8 is enriched at centrosomes. The authors also found that the distribution of gamma-tubulin and of PCM1 was scattered in the progeny of Wdr8^{-/-} mothers, instead of being present in a focus as in the wild-type. These observations led the authors to conclude that Wdr8 is required for the maintenance of functional MTOCs during embryonic mitoses. Additional experiments revealed that EYFP-tagged WD40 repeat domain mutants do not localize to centrosomes and also do not sustain phenotypic rescue amongst the progeny of Wdr8^{-/-} homozygous mothers. The authors then tested whether the WD40 domains of Wdr8^{-/-} direct the protein to centrosomes. To this end, they fused the centrosomal localization signal PACT to Wdr8 WD40 mutants; however, this did not yield rescue. The authors then tested whether the Wdr8 WD40 domains may be important through an association with the centrosomal partner protein SSX2IP, with which it interacts in other stems. Compatible with this view, the authors conducted co-immunoprecipitation experiments, which revealed that SSX2IP interacts with wild-type Wdr8, but not the WD40 domains mutant

variants. Moreover, the authors found that Wdr8 colocalized with SSX2IP in the embryo, and that the two proteins depended on another for proper centrosomal localization.

This is an interesting study that sheds light on the mechanisms through which centrosomes assemble in the early vertebrate embryo. The work is well executed for the most part. Despite the fact that some related information was known previously (e.g. about the interaction between Wdr8 and SSX2IP in other systems), publication is recommended provided the authors address the following outstanding issues in full.

Main points:

1) Why is there a delay in cell cycle progression, in particular at first cleavage, in zygotes derived from Wdr8^{-/-} mothers? Do the authors think that this is due merely to a longer time spent in mitosis? If so, can this be bypassed by inactivating the Spindle Assembly Checkpoint? An alternative possibility is that Wdr8^{-/-} exerts centrosome-independent functions as well.

Thanks for this comment, which touches a very important point that was not fully clear in our initial submission. We have carefully analysed the timing of the first mitotic division and further evaluated our data. The revised results in Fig. 1b and Supplementary Table 1 now include data of offspring from four different mating combinations. Our analysis indicates that, in the first cleavage cycle, the average timing of WT and all crosses is uniform and stays within a four-minute time window at maximum. These results show that there is satisfaction of the spindle assembly checkpoint in the first mitotic cell cycle, in contrast to what has been described for the absence of paternal contributions that cause delay in the first cell cycle (Yabe, T., Ge, X. & Pelegri, F., *Dev. Biol.*, 2007, Supplementary Fig. 10). On the other hand, the zygotes from Wdr8^{-/-} homozygous mothers gradually began to delay their cleavage timings and increased their variability from the second cleavage cycle on, accumulating these delays in the 3rd and 4th cleavage cycle (Fig. 1b and Supplementary Table 1). Moreover, we found defects in centrosome function / spindle formation only from the second division on, but not during the first cleavage (See details in Supplementary Fig. 10). As previously reported (O'Connell, K. F. et al., *Cell*, 2001, Yabe, T., Ge, X. & Pelegri, F., *Dev. Biol.*, 2007), the centrosome assembly in the first cleavage appears to be regulated mainly by paternal centrosome (centriole) factors in medaka zygotes. After the first cleavage, this mechanism seems to switch to the regulation mediated by maternal centrosome factors such as Wdr8.

This model is also corroborated by Supplementary Fig. 4, in which the cytokinesis failure significantly arises from second cleavage of the offsprings from $Wdr8^{-/-}$ homozygous mother.

2) The authors should consider raising antibodies against medaka Wdr8. Perhaps expression of EYFP tagged HuWdr8 does not faithfully mimic that of the endogenous protein (despite the observed rescue), in particular if expression levels differ and/or if the presence of the tag perturbs localization. For instance, it may be that the centrosomal enrichment is observed only upon overexpression of the protein.

We have raised and tested numerous antibodies against medaka Wdr8 to enable us to detect the endogenous protein both by Western blot and immunofluorescence. While these antibodies have worked in ELISA and detected recombinant antigens in Western blots, we were, unfortunately, unable to detect the endogenous protein by Western blot or immunofluorescence.

Of note, the EYFP-tag has been used in huWdr8 in a recent publication reflecting the centrosomal localisation in cultured cells (Kurtulmus, B. et al., *J. Cell Sci.*, 2016). Here, overexpression does not seem to contribute to the forceful enrichment to the centrosome. In our experiments, even EYFP-huWdr8 from very low concentration of mRNA (5 ng/ μ l), which cannot rescue $Wdr8^{-/-}$ zygotes, efficiently localised to the centrosome, suggesting that centrosomal Wdr8 is not targeted due to overexpression of EYFP-Wdr8 (Please see Supplementary Fig. 8, and lines 173-175). In addition, the high sequence conservation (see Supplementary Fig. 1) together with previously published data on the centrosomal localisation of human Wdr8 (Hori, A. et al., *Biochem. Biophys. Res. Commun.*, 2015 Kurtulmus, B. et al., *J. Cell Sci.*, 2016) strongly suggest that Wdr8 acts as a centrosomal component in Medaka as well.

3) Related to the above comment, how does the amount of injected EYFP-HuWdr8 compare with those of endogenous Wdr8? At the minimum quantitative RT-PCR experiments should be conducted to clarify this point.

We have performed RT-PCR to determine the level of exogenous EYFP-huWdr8 relative to endogenous Wdr8 transcript. The results are presented as Supplementary Fig. 9. In order to achieve a relative quantification, the cDNAs of WT, $Wdr8^{-/-}$, and rescued $Wdr8^{-/-}$ at 80mpf were utilised. At 80mpf, corresponding to the end of the first or the entry into the second mitosis (Fig. 1b), the centrosome/spindle abnormalities are first obvious in $Wdr8^{-/-}$ zygotes (See 60-70mpf in Supplementary Fig. 10, please note that

due to incubation at 30 °C, the cleavage cycles were faster than those at 28 °C (in all other experiments).). Therefore, 80mpf defines the critical time to rescue *Wdr8*^{-/-} zygotes. At this time, rescuing exogenous EYFP-huWdr8 mRNA is at a 45-fold excess compared to endogenous *Wdr8* mRNA. The non-availability of functional antibodies so far prevented addressing the actual excess on the level of the protein. It should be noted that in medaka experiments, concentrations between 50-500 ng/μl mRNA are usually preferred for microinjection into 1-cell stage zygotes (Kinoshita, M et al., *Medaka: Biology, Management, and Experimental protocols*, Wiley-Blackwell, 2009). Therefore, the concentration used here (100 ng/μl of mRNA) is rather at the lower end of concentrations that were used in the past to observe physiological expression and localisation.

For the full evaluation of active levels one needs to consider the following parameters:

1. The efficiency of protein translation from the exogenous mRNA might be lower than from endogenous mRNA due to different composition of 3'UTR such as poly-A signal etc.
2. The biological activity of human *Wdr8* fused to EYFP may be lower than that of the endogenous medaka *Wdr8* protein.
3. Since exogenous *Wdr8* requires a certain time for the protein translation and maturation (Note that endogenous maternal *Wdr8* is already translated and is folded properly in oocytes), rather high amount of exogenous mRNA must be introduced to swiftly reach the sufficient protein level to rescue *Wdr8*^{-/-} zygote.

4) Besides testing the PACT domain as a means to target WD40 mutant variants to the centrosome, the authors should consider testing also the centrosomal localization signals from cyclin E (CLS; PMID: 15514162) and that encoded by exon 9 of human *Sgo1* (CTS; see PMID: 26365192). Fusion to the PACT sequence with the WD40 mutant variants resulted in only weak centrosomal localizatio, but perhaps stronger localization to centrosomes achieved by the CLS or the CTS could lead to more substantial rescue.

This is a very good suggestion: we have used both the CTS and the CLS sequences to target *Wdr8* mutant variants (196/197AA) to centrosomes. Unfortunately, neither of the two alternative centrosomal targeting sequences have worked to promote centrosomal localisation (please see attached data as below). Consistently, they were unable to rescue the cell division defects observed in *Wdr8*^{-/-} zygotes. Therefore, we decided to show only the PACT fusion data in Fig.5a.

Data 2 | Centrosomal targeting of the EYFP-WD mutant variant (196/197AA) with various centrosomal targeting motifs to rescue $Wdr8^{-/-}$ zygotes. External phenotypes (a) and rescue efficiency (b) of $Wdr8^{-/-}$ zygotes injected with either EYFP-196/197AA-CLS or EYFP-196/197AA-CTS. Neither of the centrosomal targeting motifs was able to localise EYFP-196/197AA to the centrosome although fluorescence was readily detectable in injected zygotes (2.6hpf, 8 cell in the rescue). As a result, at 1.5dpf, neither EYFP-196/197AA-CLS nor EYFP-196/197AA-CTS were able to rescue $Wdr8^{-/-}$ zygotes in contrast to wt EYFP-huWDR8 (a, b, and Fig.5a). Scale bars, 200 μ m. n, total number of zygotes from three independent experiments.

5) The authors assert that EYFP-HuWdr8 and SSX2IP co-localize, supporting the notion that they are part of a complex. Conventional confocal microscopy is not adequate to determine this -a super-resolution approach would be needed to reach this conclusion with certainty.

We agree with this formal argument. Our efforts to detect Wdr8 and SSX2IP by super-resolution microscopy (two-colour STED) in

medaka have, however, failed due to technical limitations (signal to noise ratio, sample thickness). We refer to previously published data (Hori, A. et al., *Biochem. Biophys. Res. Commun.*, 2015 Kurtulmus, B. et al., *J. Cell Sci.*, 2016), in which superresolution microscopy has been used to dissect the localisations of SSX2IP and Wdr8 in cell culture. Together with our data on the interaction of Wdr8 and SSX2IP using biochemical assays, and the high conservation of the interaction, we still believe that there is strong evidence for such a complex in fish embryos.

6) The authors propose that the WD40 domains of Wdr8 mediate complex formation with SSX2IP, and thus drag the latter protein to centrosomes. If this is true, then fusing the WD40 domain mutants to SSX2IP should rescue the phenotype incurred following loss of Wdr8 function. Has this experiment been attempted?

We have injected an mRNA encoding SSX2IP fused to a WD domain mutant (EYFP-196/197AA). However, we did neither observed a rescue of the *Wdr8*^{-/-} phenotype (see attached data as below) using an N- or C-terminal fusion of EYFP-196/197AA. We hypothesise that Wdr8 is required not only as a receptor for SSX2IP but needs to fulfil functions in the complex. We have therefore toned down our original statement and now phrase the following: "*In summary, our results demonstrate that maternal Wdr8 is essential for PCM assembly during the rapid mitoses in early embryonic development. It further suggests that Wdr8 conducts its functions in a complex with SSX2IP.*" (p.12, lines 286-288).

Data 3 | Centrosomal targeting of EYFP-WD mutant variant (196/197AA) by fusing SSX2IP to rescue $Wdr8^{-/-}$ zygotes. External phenotypes (a) and rescue efficiency (b) of $Wdr8^{-/-}$ zygotes injected with either EYFP-huWdr8 or EYFP-196/197AA-SSX2IP. EYFP-196/197AA-SSX2IP was able to localise to the centrosome (2.6hpf, 8 cell in the rescue), although weaker than EYFP-huWdr8, but failed to rescue $Wdr8^{-/-}$ zygotes (1.5dpf) (a, b). Scale bars, 200 μ m. n, total number of zygotes from three independent experiments.

7) In many place, the number of embryos examined is missing; stating "all embryos" (e.g. on l. 113) without stating the number of embryos analyzed is insufficient. Likewise, numbers must be given to qualify statements such as "... failure of cytokinesis... which was sometimes observed".

The numbers of embryos used in every individual experiment are now indicated whenever needed for a quantitative statement, i.e. in Fig. 1b and Supplementary Fig. 4.

Smaller points:

8) Since the authors use human Wdr8 to test function in medaka, and then the human protein to test interactions in *Xenopus* extracts, they should provide a Supplementary Figure to show the relatedness of these proteins (i.e. Wdr8 and SSX2IP proteins).

A phylogenetic tree and an alignment of Wdr8 sequences among vertebrates have been added to Supplementary Fig. 1.

9) The authors should test whether the localization of EYFP-HuWdr8 around centrosomes is dependent on intact microtubules.

We have used nocodazole to inhibit microtubule polymerisation. Interestingly, this abolished centrosomal localisation. We therefore indeed conclude that Wdr8 localises to centrosomes in a microtubule-dependent manner. The result is now, as Fig. 2d, a part of the results section.

10) It is puzzling that Wdr8 should not be required for PCM assembly in somatic cells, because such a requirement also exists in human somatic cells (reference 34 quoted in the manuscript under consideration here). The authors might want to comment a bit more on this point.

We appreciate this important conceptual comment and have added the following paragraph to the discussion: page 13, lines 303-309, *"This differs from a certain dispensability of centrosomes in somatic cells in cell culture^{7, 8}. It seems likely that functional requirements for centrosomes in rapidly dividing vertebrate zygotes are much more strict than in somatic cells. While complete loss of functions of centrosomal proteins can be compensated in single cells in culture, multicellular cell arrangements of embryos fail to do so. This even more underlines the importance of exploiting in vivo models for the analysis of centrosomal proteins in cell division."*

11) On numerous occasions, the authors state that their findings "clearly demonstrate" this or that. It appears that writing "demonstrate" is more than enough.

We have toned down our statements throughout the manuscript in this respect.

Minor points/typos:

- l. 155-159: awkward sentence; please clarify/polish corrected, now in lines 169-176.

- l. 181-183: awkward sentence; please clarify/polish corrected, now in lines 210-213.
- l. 272-274: somewhat strange concluding sentence; please clarify/polish corrected, in lines 310-311.
- l. 448: incomplete sentence corrected, now in line 571.
- In the legend of Fig. 1a, please spell out what the stippled areas represent. corrected, please see the legend of Fig.1a.
- Does Fig.1b truly report minima and maxima?
Boxes and whiskers represent median with 25th and 75th percentile and maximum/minimum, respectively. Please see the revised Fig.1b and its legend and Supplementary Table 1.
- l. 651, "by contrast" is meant instead of "contrary". corrected, now in line 818.

Reviewer #3 (Remarks to the Author):

Inoue et al characterize the function of an identified maternally contributed centrosomal protein, Wdr8, in the early medaka embryo. They generate a CRISPR/Cas9 allele in this gene and find that absence of maternal Wdr8 function leads to defects in cell division, such as aberrant and delayed cleavage followed by aborted embryogenesis. They also begin to characterize the interactions between Wdr8 and other proteins involved in centrosomal assembly, finding that Wdr8 binds SSX2IP as well as gamma tubulin and PCM1. They are able to express functioning Wdr8 protein in the early embryo and, using this assay and in vitro mutagenized proteins, address the role of the WD40 domains in Wdr8, and find that the SSX2IP interaction is dependent on the integrity of this domain yet that with gamma-tubulin and PCM1 is not. Together with localization of the endogenous and expressed proteins, the manuscript presents a model for the localization of these proteins in centrosomes as they function in centrosome assembly.

Major

- The presentation of the parental vs zygotic effects of Wdr8 loss of function does not clearly state whether there is a paternal contribution (independent of a maternal one), which may lead to a one-cell cycle delay in the first cell cycle, as shown in *C. elegans* and zebrafish for mutations in centriolar components. Images of embryos from mutant fathers are shown in Fig 1a but there is no timeline so that it is not possible to assess this. Such a delay would result in tetraploid embryos. Embryos from Wdr8 mutant fathers are shown to be normal in Supp Fig. 2, so presumably there is no delay (because tetraploid embryos may not be viable), however, the timing of the cell division is the primary assay in the manuscript and this is not clearly shown. Given the behavior of centriolar genes,

specifically addressing this would be important. It would be interesting to compare to the behavior of paternal mutations in centriolar genes (delay in first cell cycle) and centrosomal genes (presumably no delay, as reported in this manuscript).

This is a very important point. We have carefully analysed the respective contributions of paternal vs. maternal Wdr8 in a large number of zygotes; these data are now summarized in Fig. 1b, Supplementary Table 1, Supplementary Fig, 4, and Supplementary Fig, 10. In summary, apparent one cell-cycle delay in both paternal (from WT ♀/Wdr8^{-/-} ♂) and maternal Wdr8^{-/-} zygotes (from Wdr8^{-/-} ♀/ WT ♂ and m/z Wdr8^{-/-}) could not be seen. Over 90% of zygotes of maternal and paternal Wdr8^{-/-} zygotes enter the 1st cleavage within 90 min with about 6 min variability at most as do Wt, which is not comparable to a previously described one cell-cycle delay (25 min delay) due to the absence of paternal contribution (please check Fig.7A-D of Yabe, T. et al, *Dev. Biol.*, 2007). The zygotes from Wdr8^{-/-} homozygous mothers gradually began to delay their cleavage timings and increased their variability from the second cleavage cycle on, accumulating these delays in the 3rd and 4th cleavage cycle (Fig. 1b and Supplementary Table 1). Supplementary Fig. 4 and Supplementary Fig. 10 now corroborate the above result and further show significantly increased abnormalities in the centrosome and spindle assemblies, and cytokinesis failure from the second mitosis on. Interestingly, as this reviewer suggested, we also confirmed that Wdr8-related centrosomal defects were uncoupled from cell cycle progression (Supplementary Fig. 10a, b). These lines of evidence strongly suggest that, as previously reported (O'Connell, K. F. et al., *Cell*, 2001, Yabe, T., Ge, X. & Pelegri, F., *Dev. Biol.*, 2007), the centrosome assembly in the first cleavage division may be regulated by paternal/maternal centrosome (centriole) factors in medaka zygotes. In turn, after the first cleavage, this mechanism seems to switch to the regulation mediated by maternal centrosome (PCM/CS) factors such as Wdr8. We have integrated this interpretation into our discussion with the following statement: "Our CRISPR-Cas9 knockout medaka line also segregates the distinct regulation of centrosome assembly by paternal or maternal contribution mechanism in the first and from the second cleavage on (Supplementary Fig. 10). In general, paternal and maternal centriole factors will enable the first cell division to proceed normally but maternal centrosome (PCM/CS) factors will become even more important from the second cell cycle onwards (Supplementary Fig. 10). We speculate that in m/zWdr8^{-/-} zygotes male centrioles are functional despite the final absence of Wdr8 and still organise proper MTOC in the very first mitosis^{10, 14, 17}. Maternal Wdr8, however, becomes critical for centrosomes assembly and function from the second mitosis on. Given that the

presence of Wdr8 during early embryogenesis in our rescued animals is sufficient to allow development to adult stages, this may indicate that Wdr8 functions to “seed” centrosome formation during early development but may be expendable later. Other factors, such as Sas-6¹⁷ may maintain the integrity of centrioles and centrosomes, e.g. in male reproductive cells. Although shared paternal and maternal contributions in centrosome formation seem to be conserved across species and thus biologically important^{10, 14, 17}, how and why the zygote has established them remains largely unknown.” (p.13/14, lines 312-328).

- Why do Wdr8 maternal mutants exhibit defects in cell cleavage (furrow positioning?) This phenotype is not addressed in the manuscript. Models of cell cleavage positioning have been proposed (Wühr et al, 2010) and such models may help explain the observed phenotype.

We favour the explanation that cytokinesis defects arise due to defects in spindle assembly resulting in chromosome segregation errors; both spindle assembly defects (multipolar spindles) and cytokinesis failure were frequently observed in m/z Wdr8^{-/-} zygotes. We have now also addressed the latter point in a quantitative manner (Supplementary Fig. 4). Although Wühr et al. indicate that the interphase astral microtubule defines the position of cleavage furrows prior to mitosis, we observed the clear MTOC dispersion from second mitosis (Supplementary Fig. 10a, b). It is still possible that interphase MTOC is also affected in Wdr8^{-/-} since we observed some abnormal positioning of MTOC in the second S phase (Supplementary Fig. 10a, 60'). Therefore, we consider that both chromosome segregation errors by MTOC defects and abnormal MTOC positioning in interphase could synergistically affect furrow positioning.

- The proposed delay in cell cleavage cycle in Wdr8 maternal mutant embryos contrasts with the general behavior of centriolar defects, in which cell cycles appear to occur synchronously, but are skipped if the cell cycle is defective. In particular Figure 1b, which compares the timing of cleavage in wild type and maternal Wdr8 mutant embryos shows: a) a delay in the cycling and b) an increase in variance. One concern is that the delay in cycling is a consequence of a possible arrest of the first cell cycle if the fathers are mutant (which is not possible to discern from the figure legend), so that the first cell division in mutants occurs at the time of the first cell division in wild type and so on.

We also appreciate this comment. Please refer to the first point in

the rebuttal to this reviewer (3): We have carefully analysed the respective contributions of paternal vs. maternal Wdr8; these data are now summarized in Fig. 1b, Supplementary Table 1, Supplementary Fig. 4, and Supplementary Fig. 10. We also present data in Supplementary Fig. 10 on the relationship between the cell cycle and the centrosome defects, in which m/z Wdr8^{-/-} zygotes showed comparable progression in the first cell cycle as in WT.

Another concern is that the increase in variance could be caused by having a smaller pool of mutant embryos to study, which may have artefactually led to more asynchronous data. The manuscript methods do not clearly state how embryos were synchronized so that this is also difficult to assess (the subsection of the methods that addresses Rescue states that "eggs were immediately collected after checking mating" - one concern is that if homozygous females lay eggs more irregularly that may increase cell division variance. IVF would be the more safe-proof way to insure synchronicity. It is also not clear if the data IN Fig. 1B is from embryos which are individually followed through time (which could independently confirm that indeed there is an increase in variance in cell cycle duration) or as a group (where variance could be due to initial asynchronicity in fertilization).

This is certainly a valid point. We have carefully analysed the variance of cleavage division by collecting more samples for the experiments in Fig.1b and others, and state about the synchronisation of the zygote development in the methods section as follows: "In order to avoid unsynchronised development from different clutches of zygotes, individual adult medaka couples were independently mated and checked for spawning. The individual clutches of zygotes were collected immediately after spawning, and then independently incubated at 28 °C (except for Supplementary Fig. 10) or 30°C (Supplementary Fig. 10). Particularly, for the rescue experiments with various mRNA constructs, injection of the mRNA was carried out within 5-10 min post fertilisation in precooled 0.5x ERM medium to slow down development. 5-100 ng/μl mRNAs encoding individual constructs were injected into Wdr8^{-/-} zygotes at about 1/3 volume of the one-cell stage. The zygotes were incubated in 0.5x ERM at 28 °C. Embryos at 4 hpf (St. 8 in WT) were fixed with 4% PFA in 1x PTw to perform whole mount fluorescent immunostainings. Images of external phenotypes of live embryos were taken under Nikon SMZ18 binocular microscope with the NIS-Elements F4.00.00 imaging software (Nikon)" (p.21/22, lines 520-533).

Regarding the IVF system to synchronise zygote development, IVF of medaka is not practical to achieve swift and repeatable rescue experiments with Wdr8 mutant variants. IVF will affect on

the viability of female medaka, and female medaka requires a rest for at least two weeks to recover (Kinoshita, M et al., *Medaka: Biology, Management, and Experimental protocols*, Wiley-Blackwell, 2009). Considering that $Wdr8^{-/-}$ fish are a precious resource, these factors will be drawbacks to conduct experiments in a limited period of time. Therefore, we took the above procedure for the synchronisation.

- Related to this, why is there no variance for the 1st and 4th divisions for the mutant in Figure 1b?

We concede that the analysis of WT vs. $m/zWdr8^{-/-}$ zygotes in Fig.1b lacked a thorough quantification. Reviewer 3 indicated this point also when asking for a dissection of paternal and maternal contributions. We are of the opinion that the revised quantification of the timing of cleavage cycle with a much larger sample size allows us to draw robust conclusions (Fig. 1b and Supplementary Table 1). While we don't see any difference in the first mitotic division (Fig. 1b and Supplementary Table 1), broader variance and delay arise from the second cleavage and accumulate in the following cycles. In the graph of Fig.1b, boxes and whiskers represent median with 25th and 75th percentile and maximum/minimum, respectively. Please see the revised Fig.1b and its legend and Supplementary Table 1.

- Fusions to the PACT centrosome targeting domain are used to localize the WD mutant proteins to the centrosome. The lethality is not rescued and the authors show that localization to centrosomes is only weak. The fact that the proteins are (to some extent) localized to the centrosomes leads the authors to conclude that the WD40 domains are not required for only centrosome targeting. However, function could require a threshold level of localization. In addition, the authors subsequently show that WD40 domains interact with SSX2IP - could it be that that interaction is mediating $Wdr8$ centrosome localization? Perhaps the wording could be modified here so that the logic does not appear to have these caveats.

This is another valid point (see also comments and data (Data 3) to the criticism of reviewer 2): We have injected an mRNA encoding SSX2IP fused to a WD mutant variant. However, we did not observe a rescue of the $Wdr8^{-/-}$ phenotype. Based on these results, we hypothesise that $Wdr8$ is required not only as a receptor for SSX2IP i.e. to efficiently localise SSX2IP to the centrosome over threshold level, but needs to be present in the complex even after the centrosomal localisation. We therefore now phrase as follows: "This failure – even when $Wdr8$ was localised to the centrosome –

strongly suggests that a threshold level of centrosomal Wdr8 is critical for Wdr8 activity, for which WD40 domains specifically mediates centrosomal targeting." (p.11, lines 254-257)

- The Discussion states that rescued Wdr8 animals grow to be fertile adults. The fact that these adults are fertile is important for the reasons outlined above (on potential cell division lags and ploidy duplication) and the information should be presented in the Results and not mentioned in the Discussion without having previously presented data.

Based on the revised data, we have not observed significant potential cell division lags and paternal contribution in the first cleavage division. Likewise, neither rescued nor paternal Wdr8^{-/-} zygotes exhibited any heart edema and the lack of swim bladder inflation as paternally cea (sas-6) zebrafish mutant (please check Fig.7F of Yabe, T. et al, Dev. Biol, 2007).

The rescued Wdr8^{-/-} or zygotic Wdr8^{-/-} adults are currently being investigated as an ongoing project to reveal function of zygotic Wdr8 such as in ciliogenesis (see discussion, lines 397-403). So far, we have found that zygotic Wdr8 are expressed in the brain but absent in Wdr8^{-/-} adults. However, there were no cell cycle delay and proliferation defects in Wdr8^{-/-} adult fish by immunohistochemistry and BrdU experiments, suggesting that even in the adult there is no obvious cell-cycle delay. Therefore, we are currently investigating additional functions of zygotic Wdr8 in these rescued Wdr8^{-/-} fish. We consider that presenting some preliminary results in the manuscript will be premature and distract from our finding on the centrosomal function of maternal Wdr8 in early embryonic mitosis.

- Number of embryos tested is not clearly specified in all experiments (text or figures)

The numbers of embryos used in every individual experiment are now indicated whenever needed for a quantitative statement in the text and figures, i.e. in Fig. 1b and Supplementary Fig. 4.

Minor

Line 45: eliminating centrosomes in oocytes seems necessary to prevent "abnormal embryonic mitosis that would produce excess number of centrosomes". The logic here seems to be reversed, as elimination of maternal centrioles are important to prevent excess centrosomes in the early embryo, which would lead to abnormal mitoses corrected to "...abnormal embryonic mitoses that would be a consequence of an excess number of centrosomes"(p.2, lines 45-46).

line 78: typo "embryogenesis" corrected now in line 80.
line 306: "the localization of SSX2IP".... localization to centrosomes? corrected to "at centrosomes" (p.15, line 363).

The manuscript wording at times appears to generate overstatements and would be improved if toned down
lines 46-47: "a pool of so far unidentified maternal factors" (many factors are known in various model systems involved in centriolar and centrosome assembly in the embryo) corrected: "...maternal factors that have been characterized to some extent but not been comprehensively identified." (p.2, lines 48-49).

lines 264-267: this statement as written seems to undermine the value of forward genetic efforts, and clearly forward and reverse approaches have different strengths and weaknesses. Perhaps a more even-handed way of presenting the advantage of the author's approach is the fact that after targeting a specific gene, the authors can recapitulate the rescue of the event through RNA injection immediately after fertilization. This ability in medaka does provide an undoubted advantage compared to other systems where injection of exogenous RNA may not be effective due to a time delay of translation from injected mRNA. It would be interesting if the authors could comment on this difference and why they think such a fast rescue is possible in medaka (for example, does this approach work in medaka because they are slower dividing cells?). Would this ability to rescue be expected to be a general phenomenon of medaka (as suggested in lines 288-289) or a peculiarity of Wdr8?

We agree with the reviewers' point and suggestion. We see that the observed rescue efficiency could be due to the special feature of the medaka system (1,2), or due to the properties of Wdr8 (3) or both: 1. Slower development of medaka than that of zebrafish could give time to translate and mature exogenous proteins. 2. Maternal mRNAs may have unknown specific sequences other than 3'UTR to stabilise or increase translational efficacy (Mishima, Y. & Tomari, Y., *Mol. Cell*, 2016). Another plausible explanation (3) is that the requirement of Wdr8 from the second cleavage cycle on gives a certain time to sufficiently accumulate the protein and to rescue Wdr8^{-/-} zygotes. If Wdr8 was required from the first cell cycle on, the rescue could have been more difficult. Since this is the first study demonstrating efficient rescue after CRISPR-Cas9 knock-out in medaka, however, we do not know if this high rescue efficiency is a general phenomenon of this species. The above discussion is now in page 14, lines 332-338.

Lines 268-269: "we for the first time reveal the detailed molecular

mechanism of centrosome assembly". This should be toned down, considering the multitude of factors involved in centrosome assembly and that the manuscript addresses the function of a small subset of these factors. This has been corrected: "In this study, we established a robust *in vivo* rescue approach to reveal detailed molecular mechanisms of centrosome assembly in early vertebrates." (p.13, lines 299-300).

Previous studies in various model systems have shown chromosome segregation errors when centrioles are misregulated and it would be appropriate if the manuscripts discusses such studies and relates findings (and novelty of) to those prior studies.

We appreciate this comment. The discussion related to the previous studies is now in page 16, lines 384-393.

Reviewers' Comments:

Reviewer #1 (Remarks to the Author)

The revised manuscript have satisfactorily addressed my concerns.

Reviewer #3 (Remarks to the Author)

Review of revised manuscript NCOMMS-16-01525A by Inoue et al.

The authors conducted experiments and altered text and figures to address most of the concerns raised by the three reviewers of the original submission. The manuscript is strengthened as a result. However, a few outstanding issues remain to be addressed before publication could be recommended, as explained below.

Main points:

1) In response to comment #6 of my initial review, the authors conducted an experiment in which they fused the WD40 domain mutant version of Wdr8 to SSX2IP. They observed that this fusion protein does not rescue the phenotype of *m/zWdr8*^{-/-} embryos; this is shown as "Data 3" in the rebuttal letter. It appears from the image show therein that this fusion protein does localize to centrosomes, in a manner that may be analogous to that of the wild-type EYFP-huWdr8 counterpart, eyeballing the ratio between centrosomal and cytoplasmic signals ratios in this single image. The authors should quantify and compare these ratios with those in embryos expressing the wild-type version of the protein. Overall, this is a very important piece of data that must be shown in the manuscript, as well as discussed thoroughly, since it appears to contradict the model put forth by the authors according to which the WD40 domains of Wdr8 are critical for the localization of SSX2IP to centrosomes.

2) The authors rationalize (e.g. on page 13 of the manuscript or in response to the first major comment of reviewer 3) that the lack of apparent defect in the first cell cycle may be due to a paternal contribution of Wdr8, despite the fact that this is the case also in zygotes derived from maternal and paternal Wdr8^{-/-} animals. Furthermore, the authors suggest that there is evidence in other systems that centrosome assembly in the first cell cycle is regulated also paternally, mentioning for instance the work from O'Connell et al. 2001 in *C. elegans*. My understanding regarding the situation in the worm is different. Whereas there is indeed a paternal contribution for centriole formation, as shown in the O'Connell et al. 2001 study, PCM assembly in the first cell cycle is strictly under maternal control, as evidenced by the fact that maternal-only depletion of PCM assembly factors such as *spd-2* or *spd-5* yields a fully penetrant centrosome assembly defect in the first cell cycle already. The authors must revise their interpretation of why Wdr8 depletion does not affect the first cell cycle in medaka in the light of these considerations.

Other points:

- The fact that fusing the WD40 domain mutant version of Wdr8 to either CLS or CTS also does not result in centrosomal localization or in rescue (currently shown only as "Data 2" in the rebuttal) should be reported in the manuscript, perhaps as a Supplementary Figure, and discussed, also in the light of the first main comment above.
- Legend of Fig. 3. The fraction of embryos exhibiting multipolar spindles should be indicated.
- Line 66: the authors mention "special regulatory cues for zygotic centrosome assembly". What exactly are they referring to? This should be clarified.

- Line 165: stating that EYFP-HuWdr8 and gamma-tubulin "colocalize" based on the evidence presented in Fig. 2c is a stretch. In fact, EYFP-HuWdr8 seems to co-localize with PCM1, not gamma-tubulin, with which it merely overlaps. Also, higher magnification views would be a welcome addition to this figure to render these points in a clearer fashion.
- Line 176: the statement that "Wdr8 is involved in centrosome assembly" is premature at this point of the manuscript, since the data supporting this conclusion is shown in Fig. 3, which is introduced in the section that follows.
- Line 203: please clarify the part of the sentence that reads "...indicating that the cell cycle was synchronized despite the centrosome defect." Synchronized with what?
- Line 323: the authors probably meant to write "dispensable", not "expendable".
- Line 327: the part of the sentence that reads "... how and why the zygote has established them remains largely unknown" is unclear. Do the authors mean how this was acted upon by natural selection? Please clarify the thinking and alter the wording accordingly.

Reviewer #4 (Remarks to the Author)

In this revision, Inoue et al largely address previous reviews, in particular concerns on the necessary analysis discriminating paternal vs. maternal contribution on the phenotype, concerns about egg synchronization and analysis of timing variability, and overall writing of the manuscript. Below are some minor issues that should be considered to further improve the manuscript:

- On synchronizing clutches: a) It is still not entirely clear how authors synchronize their embryos. Authors state that zygotes are "collected immediately after spawning". How many minutes are allowed for spawning? How many minutes after spawning are embryos collected? (what does "immediately" mean?). b) When embryos are injected and transferred to precooled ERM medium, then transferred back to 28 C, how much time were embryos in cooled medium?
- If there is asynchrony in the timing of cleavage formation, it is unclear how authors determine the timing of multiple furrows that may be forming in the same embryo during the same cell cycle (e.g. each of the multiple furrows formed during the third and fourth cell cycles). Would each furrow be measured separately?
- The authors nicely show that the timing of the cell cycle is normal in Wdr8^{-/-} mutant embryos using the cycling of DNA, which fits with previous studies on defects in centriole/centrosome formation not affecting the nuclear cycle. This raises the question of why there is variability in the timing of furrow formation, which one would expect would be linked to the DNA cycle. Could it be that furrow formation timing appears to be more asynchronous than wild type because of asymmetry in membrane deposition caused by the cleavage positioning defects (e.g. membrane at the furrows being added at different rates due to placement of the furrows, leading to the appearance of furrowing at different rates)?
- Figure 1 legend, panels c,d, please specify full genotype with respect to maternal and paternal contribution (e.g. in lines 760 and 765 whether embryos are derived from crosses with mutant or wild-type males)
- Lines 805 and 806: suggest replacing "both mutant variants' expressions were" to "expression of either mutant variant was"
- Supplementary Fig. 3: please clarify maternal/zygotic phenotype. Are these zygotically mutant Wdr8^{-/-} derived from heterozygous mothers?
- Supplementary Fig. 4: symbols in graph do not seem to match symbol key (e.g. solid downward triangles in graph seem to be represented by hollow upward triangles in key). Also, symbols are very close together making it difficult to discern their shape. It would help to clarify the graph further, e.g. with brackets along the X-axis binning the pairs of data columns to a genotype.
- Also in this Supplementary Fig. 4, is each data point the fraction of cytokinesis failure observed in a clutch from a different female? Please clarify.
- The authors nicely show that the neither centrosome formation nor cell division are affected in the first cell cycle, and that defects in mutants begin in the second cell cycle and continue in subsequent cycles. This information is important considering the known behavior of the pair of sperm-derived centrioles, as already referenced by the authors, which previous studies in C.

C. elegans and zebrafish have shown that on their own (even without normal centriole replication during the first cell cycle) can nucleate a centrosome and a MTOC at the pole of a spindle. The behavior of the Wdr8 maternal mutant, however, does highlight that centrosome reconstitution in Wdr8 maternal mutant eggs appears to proceed normally when centrioles are provided by the father. If the maternal defect was in a downstream step of centrosome reconstitution, one may expect that centrosomes reconstituted around the sperm-derived centrioles during the first cell cycle would be abnormal, which is not the case. Instead the normality in the first cell cycle implies that the defect in Wdr8 may occur either at the level of centriole integrity (once centrioles need to be replicated during the first cell cycle) or in an upstream step of centrosome formation linking newly synthesized centrioles and centrosome assembly. This is something that perhaps could be more clearly conveyed in the text. The authors do make a statement which has a similar intent, that Wdr8 "seeds" centrosome formation, and that Wdr8 could "serve...as a platform for gamma-TuRC/Nedd1), but highlighting this conclusion in the context of the differing behavior of the paternally-inherited centrioles and the centrioles newly made in the embryo with maternal products may strengthen a logic that places Wdr8 function upstream in a centrosome formation pathway, or a link between centriole replication and centrosome nucleation.

- Line 534: "Cytokinesis failure in the zygotes from Wdr8^{-/-} mother gave rise to one and three blastomeres in the first and second cleavage divisions". Why does the first cell division cycle fail if the first spindle is normal, as shown by the authors?

Reviewer #3 (Remarks to the Author):

Review of revised manuscript NCOMMS-16-01525A by Inoue et al.

The authors conducted experiments and altered text and figures to address most of the concerns raised by the three reviewers of the original submission. The manuscript is strengthened as a result. However, a few outstanding issues remain to be addressed before publication could be recommended, as explained below.

Main points:

1) In response to comment #6 of my initial review, the authors conducted an experiment in which they fused the WD40 domain mutant version of Wdr8 to SSX2IP. They observed that this fusion protein does not rescue the phenotype of *m/zWdr8*^{-/-} embryos; this is shown as "Data 3" in the rebuttal letter. It appears from the image shown therein that this fusion protein does localize to centrosomes, in a manner that may be analogous to that of the wild-type EYFP-huWdr8 counterpart, eyeballing the ratio between centrosomal and cytoplasmic signals ratios in this single image. The authors should quantify and compare these ratios with those in embryos expressing the wild-type version of the protein. Overall, this is a very important piece of data that must be shown in the manuscript, as well as discussed thoroughly, since it appears to contradict the model put forth by the authors according to which the WD40 domains of Wdr8 are critical for the localization of SSX2IP to centrosomes.

The reviewer asked in comment #6 of the initial review: "The authors propose that the WD40 domains of Wdr8 mediate complex formation with SSX2IP, and thus drag the latter protein to centrosomes. If this is true, then fusing the WD40 domain mutants to SSX2IP should rescue the phenotype incurred following loss of Wdr8 function."

We do concede that EYFP-196/197AA-SSX2IP was unable to rescue lethality of *Wdr8*^{-/-} zygotes despite a weak but still obvious centrosomal localization. This result raised the following three possibilities: (1) The level of centrosomal accumulation of the EYFP-196/197AA-SSX2IP protein was not sufficient for the rescue as the reviewer suggested, (2) WD40 domains are critical for Wdr8 function beyond centrosome targeting, or, (3) EYFP-196/197AA-SSX2IP represents a non-functional complex due to steric problems or the loss of a dynamic protein-protein interaction.

We have initially addressed the possibility (1) by increasing the concentration of EYFP-196/197AA-SSX2IP mRNA to 300-500 ng/μl, which is a maximal level of proper mRNA injection in medaka system (Kinoshita, M et al., *Medaka: Biology, Management, and Experimental protocols*, Wiley-Blackwell, 2009). However, although the overall fluorescence intensity of our fusion protein was increased compared to the 100 ng/μl mRNA injection (Data 3a), the centrosomal accumulation of the fusion seemed not increased and no

rescue was observed (Data 3b). To further evaluate this approach and to address the functionality of the fusion construct (see possibility (3)), we generated an EYFP-huWdr8wt-SSX2IP fusion construct and tested its localization and rescue activity (Data 4a, b). Expression of EYFP-huWdr8wt-SSX2IP in *Wdr8*^{-/-} zygotes strongly increased the Centro/Cyto fluorescence ratio by a factor of 4 compared to wt *Wdr8* alone (Data 4c). However, it failed to rescue the knockout zygotes (Data 4, b). This result indicated that the *Wdr8*wt function is defective after fusing it directly to SSX2IP (We have tested both N- and C-terminal fusions, both of which did not rescue the knockout zygotes.). It is possible that the fusion proteins are not properly recognized as a cargo by dynein/dynactin motors, and/or do not recapitulate the original kinetics of centrosome and cytoplasmic transport of the *Wdr8*-SSX2IP complex. Approaches using a direct fusion of *Wdr8* and SSX2IP may therefore lead to inconsistent conclusions at least in our experimental system. We have decided that it is not worthwhile to include these data in our manuscript.

Data 3 | Centrosomal targeting of EYFP-WD mutant variant (196/197AA) by fusing SSX2IP to rescue *Wdr8*^{-/-} zygotes. External phenotypes (a) and rescue efficiency (b) of *Wdr8*^{-/-} zygotes injected with an mRNA encoding EYFP-huWdr8, EYFP-196/197AA-SSX2IP, or EYFP-huWdr8-SSX2IP. EYFP-196/197AA-SSX2IP was able to localise to the centrosome (2.6hpf, 8 cell in the rescue), although weaker than EYFP-huWdr8, but failed to rescue *Wdr8*^{-/-} zygotes (1.5dpf) (a, b). Expression of EYFP-196/197AA-SSX2IP at a higher concentration of mRNA (300 ng/μl) compromised its centrosomal localisation, resulting in developmental defect (a). EYFP-huWdr8-SSX2IP strongly localised to the centrosome but failed to rescue

Wdr8^{-/-} zygotes (a, b). Scale bars, 200 μm. n, total number of zygotes from three independent experiments. c, Centrosome vs cytoplasmic ratio of fluorescent signal of individual fusion proteins in the rescued zygotes. n, total number of blastomeres measured from four independent zygotes. The significances were analysed by Dunnett's multiple comparison test and scored as follows: **; P ≤ 0.01. The Data represent mean ± sd (b, c).

2) The authors rationalize (e.g. on page 13 of the manuscript or in response to the first major comment of reviewer 3) that the lack of apparent defect in the first cell cycle may be due to a paternal contribution of Wdr8, despite the fact that this is the case also in zygotes derived from maternal and paternal Wdr8^{-/-} animals. Furthermore, the authors suggest that there is evidence in other systems that centrosome assembly in the first cell cycle is regulated also paternally, mentioning for instance the work from O'Connell et al. 2001 in *C. elegans*. My understanding regarding the situation in the worm is different. Whereas there is indeed a paternal contribution for centriole formation, as shown in the O'Connell et al. 2001 study, PCM assembly in the first cell cycle is strictly under maternal control, as evidenced by the fact that maternal-only depletion of PCM assembly factors such as spd-2 or spd-5 yields a fully penetrant centrosome assembly defect in the first cell cycle already. The authors must revise their interpretation of why Wdr8 depletion does not affect the first cell cycle in medaka in the light of these considerations.

We appreciate this comment. In our experiments, sperm lacking Wdr8 (in m/zWdr8^{-/-} or the zygotes from Wdr8 knockout homozygous fathers) are fertile, i.e. they can promote fertilization and seem to successfully initiate first bipolar spindle formation after fertilization. This means that sperm basal bodies without Wdr8 still maintain the basic structure of the centrioles. They prevail their function as basal bodies for the sperm flagellum and to organize two spindle poles after fertilisation. Problems only arise once new centrosomes have to be produced.

O'Connell and colleagues had previously observed in *C. elegans* that knockout of zyg1 kinase as the main regulator of centriole duplication still allowed the production of gametes that fertilize but that these fail to proceed in development already during the first cell cycle. Using additional evidence from ultrastructural analysis of centrioles in zygotes, they concluded that the sperm, which developed in worms without zygotic (paternal) zyg1 contained basal bodies with a single centriole (instead of two in wt). Indeed, wt sperm allowed the formation of a bipolar spindle from two single centrioles in the first cell cycle when fertilizing eggs from zyg1 knockout mothers.

Given analogous principles in medaka, we speculate that the sperm of Wdr8^{-/-} fathers still contain two centrioles that allow the first bipolar spindle to be formed after fertilization. In addition, maternal Wdr8 may not be like an upstream PCM/centriolar components Spd-2 and Spd-5, the knockdowns and mutants of which exhibit penetrant failure of the PCM and bipolar spindle assemblies in the first cell cycle (Kemp, CA. et al., 2004, *Dev. Cell*; Pelletier, L. et al., 2004, *Curr. Biol.*). If that was the case, the PCM defect caused by the

absence of maternal Wdr8 would be partially compensated by other similar centriolar satellite/PCM proteins, which may hide the effect of Wdr8 absence in the first cell cycle but still allow the accumulation of the defect from the second cell cycle on. We agree with the reviewers that the discussion concerning this point was not specific enough and suggest to state the following – also in agreement with reviewer 4:

“Successful fertilization and bipolar spindle formation in the first mitotic cell cycle of medaka Wdr8^{-/-} zygotes indicate that the sperm from Wdr8^{-/-} fathers are functional and that they are likely to carry two centrioles, which define the two spindle poles in the first mitosis. PCM assembly seems not be acutely affected in the first cell cycle of Wdr8^{-/-} zygotes, in contrast to the immediate PCM defects upon knockdown or in mutants of the PCM components such as Spd-2 and Spd-5^{33, 34}. The fact that mitotic defects arose in our experiments from the second cell cycle onwards independently of the genotype of the father (Wdr8^{-/-} or wt) may confirm that sperm centrioles are still functional in Wdr8^{-/-} fathers, possibly due a long-lasting maternal effect during their development. Alternatively, maternal Wdr8 may not act upstream of PCM assembly, but rather secure the integrity of PCM assembly and structure. The defect of PCM assembly upon loss of maternal Wdr8 may also be partially compensated by other, functionally similar PCM/CS proteins, which may hide the effect of Wdr8 absence in the first cell cycle but still cause the accumulation of the defect from the second cell cycle on.”

We discuss the above points in Lines 318-332 in the revised manuscript and therefore left out the corresponding passage from our previous manuscript (Lines 314-319 in the previous manuscript):

“In general, paternal and maternal centriole factors will enable the first cell division to proceed normally but maternal centrosome (PCM/CS) factors will become even more important from the second cell cycle onwards. We speculate that in m/zWdr8^{-/-} zygotes male centrioles are functional despite the final absence of Wdr8 and still organise proper MTOC in the very first mitosis^{10, 14, 17}”, and in line 120 in the previous Supplementary Fig. 10, “possibly due to a paternal contribution mechanism¹⁷”.

Other points:

- The fact that fusing the WD40 domain mutant version of Wdr8 to either CLS or CTS also does not result in centrosomal localization or in rescue (currently shown only as "Data 2" in the rebuttal) should be reported in the manuscript, perhaps as a Supplementary Figure, and discussed, also in the light of the first main comment above.

Unfortunately, both motifs failed to target Wdr8 to centrosomes in our system. The experiment therefore misses to address the question that was asked, i.e. to analyze if centrosomal accumulation of Wdr8 WD40 mutants would be sufficient to rescue lethality of Wdr8 ^{-/-} zygotes. We therefore consider these results as not important enough to be included into our manuscript.

- Legend of Fig. 3. The fraction of embryos exhibiting multipolar spindles

should be indicated.

Since all the *Wdr8*^{-/-} zygotes (100%) exhibited multipolar spindles at metaphase, we have put the percentage in the legend of Fig. 3. (Line 842).

- Line 66: the authors mention "special regulatory cues for zygotic centrosome assembly". What exactly are they referring to? This should be clarified.

We clarify this sentence to "*the existence of regulatory complexes of centrosome proteins and their roles in the molecular steps for zygotic centrosome assembly*". Lines 64-65.

- Line 165: stating that EYFP-HuWdr8 and gamma-tubulin "colocalize" based on the evidence presented in Fig. 2c is a stretch. In fact, EYFP-HuWdr8 seems to co-localize with PCM1, not gamma-tubulin, with which it merely overlaps. Also, higher magnification views would be a welcome addition to this figure to render these points in a clearer fashion.

I appreciate and agree with this comment. We have changed the word in Line 165 "*colocalised with*" to "*localised next to γ -tubulin-positive PCM and co-localised with...*". We have put the magnified figures of the centrosome in Fig. 2c (please see the insets in Fig. 2c).

- Line 176: the statement that "Wdr8 is involved in centrosome assembly" is premature at this point of the manuscript, since the data supporting this conclusion is shown in Fig. 3, which is introduced in the section that follows.

As the reviewer suggested the statement has been changed to "*Wdr8 is a maternal centrosome protein which may have a role in embryonic mitoses*", Lines 176-177.

- Line 203: please clarify the part of the sentence that reads "*...indicating that the cell cycle was synchronized despite the centrosome defect.*" Synchronized with what?

We have changed the sentence to "*indicating that the embryonic cell cycles were synchronous within a population of animals, as was the case between WT zygotes*", Lines 205-206.

- Line 323: the authors probably meant to write "dispensable", not "expendable".

The sentence with this typo was removed and changed to the sentences to discuss about the reviewer's main point 2) in Lines 318-332.

- Line 327: the part of the sentence that reads "... how and why the zygote has established them remains largely unknown" is unclear. Do the authors mean how this was acted upon by natural selection? Please clarify the thinking and

alter the wording accordingly.

The wording was inappropriate at this point. We have changed the sentence to “*the molecular details of a pathway for new centrosome assembly from paternal and maternal centrosome proteins in vertebrate embryonic mitosis remains largely unknown*”, Lines 334-336.

Reviewer #4 (Remarks to the Author):

In this revision, Inoue et al largely address previous reviews, in particular concerns on the necessary analysis discriminating paternal vs. maternal contribution on the phenotype, concerns about egg synchronization and analysis of timing variability, and overall writing of the manuscript. Below are some minor issues that should be considered to further improve the manuscript:

- On synchronizing clutches: a) It is still not entirely clear how authors synchronize their embryos. Authors state that zygotes are “collected immediately after spawning”. How many minutes are allowed for spawning? How many minutes after spawning are embryos collected? (what does “immediately” mean?). b) When embryos are injected and transferred to precooled ERM medium, then transferred back to 28 C, how much time were embryos in cooled medium?

We have clarified the wording and the details for these points in the materials and methods as follows: “*The spawning of eggs was finished less than 1 min after mating. The individual clutches of zygotes were collected at 5 min after spawning (i.e. 5mpf), and then independently incubated at 28 °C (except for Supplementary Fig. 10) or 30°C (Supplementary Fig. 10). Particularly, for the rescue experiments with various mRNA constructs, collected eggs were transferred to the precooled 0.5x ERM medium at 5 min after spawning (or fertilization) to slow down development and then subjected to mRNA injection. Injection of the mRNA into one-cell stage zygotes was carried out within 10-15 min after spawning in the precooled 0.5x ERM medium. 5-100 ng/μl mRNAs encoding individual constructs were injected into *Wdr8*^{-/-} zygotes at about 1/3 volume of the one-cell stage. After injection, the precooled 0.5x ERM medium of the injection dish containing injected zygotes were exchanged to pre-warmed (28 °C) 0.5x ERM (i.e. at 10-15 min after spawning or 5-10 min after collection to the precooled medium), and the zygotes were incubated at 28 °C.*”, see Lines 531-544.

- If there is asynchrony in the timing of cleavage formation, it is unclear how authors determine the timing of multiple furrows that may be forming in the same embryo during the same cell cycle (e.g. each of the multiple furrows formed during the third and fourth cell cycles). Would each furrow be measured separately?

Since the timing of multiple cleavage furrows in the same zygote was synchronous within 1 min variability (please note that different zygotes showed different cleavage timings and thus caused higher variability of the timing in the collected data), we used the time point at which at least 50% of the blastomeres in a zygote showed furrow formation. For example, if 6 blastomeres at 8-cell stage (third cleavage) showed cleavage furrows at 153 min and the rest showed the furrows at 154 min, we indicated the time of 153 min. In turn, when a blastomere exhibited cytokinesis failure with no furrow formation (from $Wdr8^{-/-}$ ♀/WT ♂ or $Wdr8^{-/-}$ ♀/ $Wdr8^{-/-}$ ♂), we used the majority of time point that at least 50% of the rest of the blastomeres underwent cleavage furrows. For example, if one blastomere exhibited cytokinesis failure from the 3rd to 4th cleavage cycle (8-cell to 16-cell stage in WT), we counted the time point of the 7 other blastomeres undergoing cleavage divisions according to the above criterion. This measurement procedure was stated in the materials and methods, Lines 548-560.

- The authors nicely show that the timing of the cell cycle is normal in $Wdr8^{-/-}$ mutant embryos using the cycling of DNA, which fits with previous studies on defects in centriole/centrosome formation not affecting the nuclear cycle. This raises the question of why there is variability in the timing of furrow formation, which one would expect would be linked to the DNA cycle. Could it be that furrow formation timing appears to be more asynchronous than wild type because of asymmetry in membrane deposition caused by the cleavage positioning defects (e.g. membrane at the furrows being added at different rates due to placement of the furrows, leading to the appearance of furrowing at different rates)?

As in Fig. 1b, we observed that cleavage divisions in the offspring of $Wdr8^{-/-}$ mothers gradually accumulated a delaying effect. The cleavage furrow in a $Wdr8^{-/-}$ zygote was randomized but still formed throughout the entire blastomere. As noted above the cleavage timing of in the same embryo was rather synchronous in $Wdr8^{-/-}$ even though it sometimes showed cytokinesis failure. As the reviewer suggested, we consider the possibility that also membrane deposition will be affected, possibly due to furrow positioning defects. However, based on our observation, we find it more likely that the exit from the mitosis with multipolar spindles and cytokinesis failure together with chromosome segregation errors may gradually uncouple the timing of cytokinesis from the cell cycle (DNA cycle) in $Wdr8^{-/-}$. Since this consideration is, however, very speculative and not related to our focus on the specific function of $Wdr8$ for the centrosome assembly, we have not discussed this point in our manuscript.

- Figure 1 legend, panels c,d, please specify full genotype with respect to maternal and paternal contribution (e.g. in lines 760 and 765 whether embryos are derived from crosses with mutant or wild-type males)

-Lines 805 and 806: suggest replacing “both mutant variants’ expressions were” to “expression of either mutant variant was”

We have specified the genotypes of the zygotes in the legends and panels of Figure 1 c, d. (Lines 134 and 135, Lines 814-815, and Lines 819-820).
Lines 805 and 806: corrected, now in Line 861.

- Supplementary Fig. 3: please clarify maternal/zygotic phenotype. Are these zygotically mutant $Wdr8^{-/-}$ derived from heterozygous mothers?

The genotype of $Wdr8^{-/-}$ fish (from the heterozygous mother and father) was specified in Supplementary Fig. 3.

- Supplementary Fig. 4: symbols in graph do not seem to match symbol key (e.g. solid downward triangles in graph seem to be represented by hollow upward triangles in key). Also, symbols are very close together making it difficult to discern their shape. It would help to clarify the graph further, e.g. with brackets along the X-axis binning the pairs of data columns to a genotype.

We appreciate this point and suggestion. We have corrected the mismatch of the symbols in the graph. The presentation of the symbols and genotypes were also clarified.

- Also in this Supplementary Fig. 4, is each data point the fraction of cytokinesis failure observed in a clutch from a different female? Please clarify.

Each data point of the fraction of cytokinesis failure was taken from a clutch of individual different females. This is now denoted in the legend of Supplementary Fig. 4, Line 46-49.

- The authors nicely show that the neither centrosome formation nor cell division are affected in the first cell cycle, and that defects in mutants begin in the second cell cycle and continue in subsequent cycles. This information is important considering the known behavior of the pair of sperm-derived centrioles, as already referenced by the authors, which previous studies in *C. elegans* and zebrafish have shown that on their own (even without normal centriole replication during the first cell cycle) can nucleate a centrosome and a MTOC at the pole of a spindle. The behavior of the $Wdr8$ maternal mutant, however, does highlight that centrosome reconstitution in $Wdr8$ maternal mutant eggs appears to proceed normally when centrioles are provided by the father. If the maternal defect was in a downstream step of centrosome reconstitution, one may expect that centrosomes reconstituted around the sperm-derived centrioles during the first cell cycle would be abnormal, which is not the case. Instead the normality in the first cell cycle implies that the defect in $Wdr8$ may occur either at the level of centriole integrity (once centrioles need to be replicated during the first cell cycle) or in an upstream step of centrosome formation linking newly synthesized centrioles and centrosome assembly. This is something that perhaps could be more clearly conveyed in the text. The authors do make a statement which has a similar intent, that $Wdr8$ “seeds” centrosome formation, and that $Wdr8$ could “serve...as a platform for gamma-TuRC/Nedd1), but highlighting this

conclusion in the context of the differing behavior of the paternally-inherited centrioles and the centrioles newly made in the embryo with maternal products may strengthen a logic that places Wdr8 function upstream in a centrosome formation pathway, or a link between centriole replication and centrosome nucleation.

We very much appreciate this comment, the issue of which was also raised by reviewer 3. Please see our comment to the corresponding point by reviewer 3, which we line out here again:

In our experiments, sperm lacking Wdr8 (in m/zWdr8^{-/-} or the zygotes from Wdr8 knockout homozygous fathers) are fertile, i.e. they can promote fertilization and seem to successfully initiate first bipolar spindle formation after fertilization. This means that sperm basal bodies without Wdr8 still maintain the basic structure of the centrioles. They prevail their function as basal bodies for the sperm flagellum and to organize two spindle poles after fertilisation. Problems only arise once new centrosomes have to be produced. O'Connell and colleagues had previously observed in *C.elegans* that knockout of zyg1 kinase as the main regulator of centriole duplication still allowed the production of gametes that fertilize but that these fail to proceed in development already during the first cell cycle. Using additional evidence from ultrastructural analysis of centrioles in zygotes, they concluded that the sperm which developed in worms without zygotic (paternal) zyg1 contained basal bodies with a single centriole (instead of two in wt). Indeed, wt sperm allowed the formation of a bipolar spindle from two single centrioles in the first cell cycle when fertilizing eggs from zyg1 knockout mothers.

Given analogous principles in medaka, we speculate that the sperm of Wdr8^{-/-} fathers still contain two centrioles that even allow the first bipolar spindle to be formed after fertilization. In addition, maternal Wdr8 may not be like an upstream PCM/centriolar components Spd-2 and Spd-5, the knockdowns and mutants of which exhibit penetrant failure of the PCM and bipolar spindle assemblies in the first cell cycle (Kemp, CA. et al., 2004, *Dev. Cell*; Pelletier, L. et al., 2004, *Curr. Biol.*). If that was the case, the PCM defect caused by the absence of maternal Wdr8 would be partially compensated by other similar centriolar satellite/PCM proteins, which may hide the effect of Wdr8 absence in the first cell cycle but still allow the accumulation of the defect from the second cell cycle on. We agree with the reviewers that the discussion concerning this point was not specific enough and suggest to state the following – also in agreement with reviewer 3:

“Successful fertilization and bipolar spindle formation in the first mitotic cell cycle of medaka Wdr8^{-/-} zygotes indicate that the sperm from Wdr8^{-/-} fathers are functional and that they are likely to carry two centrioles, which define the two spindle poles in the first mitosis. PCM assembly seems not be acutely affected in the first cell cycle of Wdr8^{-/-} zygotes, in contrast to the immediate PCM defects upon knockdown or in mutants of the PCM components such as Spd-2 and Spd-5^{33, 34}. The fact that mitotic defects arose in our experiments from the second cell cycle onwards independently of the genotype of the father (Wdr8^{-/-} or wt) may confirm that sperm centrioles are still functional in Wdr 8^{-/-} fathers, possibly due a long-lasting maternal effect during their

development. Alternatively, maternal Wdr8 may not act upstream of PCM assembly, but rather secure the integrity of PCM assembly and structure. The defect of PCM assembly upon loss of maternal Wdr8 may also be partially compensated by other, functionally similar PCM/CS proteins, which may hide the effect of Wdr8 absence in the first cell cycle but still cause the accumulation of the defect from the second cell cycle on.

We discuss the above points in Lines 318-332 in the revised manuscript and therefore left out the corresponding passage from our previous manuscript (Lines 314-319 in the previous manuscript):

“In general, paternal and maternal centriole factors will enable the first cell division to proceed normally but maternal centrosome (PCM/CS) factors will become even more important from the second cell cycle onwards. We speculate that in m/zWdr8^{-/-} zygotes male centrioles are functional despite the final absence of Wdr8 and still organise proper MTOC in the very first mitosis^{10, 14, 17}”, and in line 120 in the previous Supplementary Fig. 10, “possibly due to a paternal contribution mechanism¹⁷”.

- Line 534: “Cytokinesis failure in the zygotes from Wdr8^{-/-} mother gave rise to one and three blastomeres in the first and second cleavage divisions”. Why does the first cell division cycle fail if the first spindle is normal, as shown by the authors?

This is partially related to this reviewer’s comment in the initial review: “- Why do Wdr8 maternal mutants exhibit defects in cell cleavage (furrow positioning?). This phenotype is not addressed in the manuscript. Models of cell cleavage positioning have been proposed (Wühr et al, 2010) and such models may help explain the observed phenotype.” In response to this comment, we noted that “It is still possible that the interphase MTOC is also affected in Wdr8^{-/-} since we observed some abnormal positioning of the MTOC in the second S phase (Supplementary Fig. 10a, 60’).” In this respect, after forming apparently normal mitotic spindles at metaphase of the first cell cycle, some of the embryos could begin to harbour abnormal positioning of the centrosome at anaphase or telophase (the time point corresponding to inbetween 50’ to 60’ in Supplementary Fig. 10a). We speculate that this abnormal positioning at the exit from the first cell cycle may cause a cytokinesis failure despite the successful assembly of the first mitotic spindles.

Reviewers' Comments:

Reviewer #3 (Remarks to the Author)

The authors have addressed in a satisfactory manner the remaining points (although I would argue that reporting both "Data 2" and "Data 3" would be in order).

Reviewer #4 (Remarks to the Author)

The authors have addressed remaining concerns through modifications of the text. In particular, explanations of how embryos were synchronized and the timing of treatment, as well as the inferred effects on paternal centriolar contribution, have been clarified.

Response to Referee's comment

Reviewer #3 (Remarks to the Author):

The authors have addressed in a satisfactory manner the remaining points (although I would argue that reporting both "Data 2" and "Data 3" would be in order).

As we have discussed in the previous rebuttal letter, we have decided not to include "Data 2" and "Data 3" for the reviewer #3's point, due to the complexity of fusion constructs and of the interpretation of the results. Reporting both data may cause readers to confuse and distract our findings with inconsistent conclusions. Please also see the details of our comment for comment #1 of this reviewer in the previous rebuttal letter.